# Monoclonal neutralizing antibodies elicited by infection with Kaposi sarcoma-associated herpesvirus reveal critical sites of vulnerability on gH/gL

Yu-Hsin Wan[1], Sara Pernikoff[1], Nicholas T. Aldridge[1], Kevin Lang[1], Holly M. Dudley[1], Samuel C. Scharffenberger[1], Gargi Kher[1], Warren Phipps[1,2], Marie Pancera[1], Jim Boonyaratanakornkit[1,2], Andrew T. McGuire [1,3,4]*

1 Vaccine and Infectious Disease Division, Fred Hutchinson Cancer Center, Seattle, Washington, United States of America, 2 Department of Medicine, University of Washington, Seattle, Washington, United States of America, 3 Department of Global Health, University of Washington, Seattle, Washington, United States of America, 4 Department of Laboratory Medicine and Pathology, University of Washington, Seattle, Washington, United States of America

* amcguire@fredhutch.org

## Abstract

Kaposi sarcoma-associated herpesvirus (KSHV) is an oncogenic virus that causes Kaposi sarcoma, primary effusion lymphoma and multicentric Castleman disease. A vaccine that prevents KSHV infection or serves in the treatment of KSHV-related diseases represents a critical unmet need, however, the types of immune responses a vaccine should elicit have not been well defined. The gH/gL glycoprotein complex is an important target of KSHV-neutralizing antibodies, but the epitope specificities targeted by these antibodies remain unknown. Here, we isolated 12 gH/gL-specific monoclonal antibodies (mAbs) from KSHV-infected donors and performed structure/ function analyses. These mAbs bind recombinant gH/gL with nanomolar affinities and epitope binning analyses revealed that the mAbs bind to 5 epitope clusters on gH/gL. Seven mAbs were able to neutralize KSHV infection of epithelial cell lines. Two potent neutralizing mAbs mapped to the EphA2 binding site as determined by inhibition of the receptor-ligand interaction and negative stain electron microscopy (nsEM) of the mAb/gH/gL complex. The epitopes of other neutralizing mAbs targeting novel sites of vulnerability were determined by a combination of cryogenic electron microscopy and nsEM. Together, these mAbs help to define the relevant epitope targets for KSHV vaccine design, have utility in understanding the role of antibodies in preventing KSHV infection, enable the development of immunotherapy approaches, and provide valuable tools to understand the molecular details of the KSHV entry process.

**Data availability statement:** All materials generated herein are available upon request under an MTA from Fred Hutchinson Cancer Center (mta@fredhutch.org). The pTT3 vectors are used under license from the National Research Council of Canada. Negative-staining EM maps are deposited in the EMDB with accession codes: EMD-72129 (Negative Stain EM map of KSHV glycoprotein gH and gL), EMD-72130 (Negative Stain EM map of KSHV glycoprotein gH/gL in complex with MLKH1 Fab), EMD-72131 (Negative Stain EM map of KSHV glycoprotein gH/gL in complex with MLKH5 Fab), EMD-72132 (Negative Stain EM map of KSHV glycoprotein gH/gL in complex with MLKH5, MLKH10 and MLKH3 Fabs), EMD-72133 (Negative Stain EM map of KSHV glycoprotein gH/gL in complex with MLKH5, MLKH10 and MLKH6 Fabs), EMD-72525 (Negative Stain EM map of KSHV glycoprotein gH/gL in complex with MLKH5, MLKH10 and MLKH12 Fabs). The cryoEM structure of the gH/gL/MLKH3/MLKH5/MLKH10 structure has been deposited in the Protein Data Bank (PDB ID 9Z3Q) and the Electron Microscopy Data Bank (EMD-73789). The sequences of the antibodies described herein are available on NCBI Genbank; accession numbers: PV035824 - PV035843 and PX122841 - PX122844. Mass spectrometry data has been uploaded to MassIVE under: MSV000099570.

**Funding:** This work was supported by the National Cancer Institute (U01 CA295050 to ATM, WP and JB, and R01 CA239593 to WP) and a Fred Hutch Vaccine and Infectious Disease Division Initiative Grant (to ATM WP and JB). This research was supported by NIH P30 CA015704 of the Fred Hutch/University of Washington/Seattle Children's Cancer Consortium, which includes the Electron Microscopy Shared Resource (RRID:SCR_022611), the Cellular Imaging Shared Resource (RRID:SCR_022609) and the Flow Cytometry Shared Resource (RRID:SCR_022613). The funders had no role in study design, data collection and analysis, decision to publish, or preparation of the manuscript. The funders had no role in study design, data collection and analysis, decision to publish, or preparation of the manuscript.

## Author summary

KSHV is an oncogenic virus that can cause cancer in infected individuals. The virus is most prevalent in sub-Saharan Africa and in men who have sex with men. It is possible this virus could be prevented with an effective vaccine, however, the immune response to this virus has not been well defined. gH/gL, a protein essential for viral fusion, plays an important role in infection and could be a possible vaccine target. To better understand the antibody response to this protein, we sought to isolate and characterize monoclonal antibodies that can bind gH/gL and neutralize viral infection. In this study, we isolate and characterize twelve monoclonal antibodies that could bind to five different regions of the gH/gL protein. Seven of these antibodies can neutralize infection, with two being able to block the gH/gL EphA2 receptor-ligand interaction.

## Introduction

KSHV is an oncogenic human gammaherpesvirus for which there is no vaccine. KSHV is the etiologic agent of Kaposi sarcoma (KS) [1], a form of multicentric Castleman (KSHV-MCD) disease [2], primary effusion lymphoma (PEL) [3] and KSHV Inflammatory Cytokine Syndrome (KICS) [4]. KSHV can be shed in saliva, which is thought to be the primary mode of transmission [5]. Two major patterns of transmission have been described [6]. In Africa, transmission largely occurs during childhood and is associated with close contact and possibly premastication of food in some cultures [7–9]. Seroprevalence is highest in sub-Saharan Africa and has been reported to exceed 80% in rural Uganda [10]. A second pattern of transmission is related to saliva exchange during sexual encounters, which is associated with an older age of seroconversion and is important in men who have sex with men (MSM) [11–13]. Outside of Africa, HIV seronegative MSM have a seroprevalence of approximately 20% [11–14].

KSHV could be a vaccine-preventable infection [15], however, it is currently unknown what types of immune responses will be important for preventing or reducing KSHV transmission, or which responses could provide therapeutic benefit in the treatment of KSHV-related diseases. The development of effective vaccines requires comprehensive information about the replication, biology, and pathogenesis of a virus, as well as an understanding of the host immune response to infection. Natural KSHV infection elicits broad antibody [16] and T-cell [17,18] responses. Neutralizing antibodies are often an important correlate of protection for effective vaccines [19,20]. Epidemiologic studies have evaluated the ability of patient sera to neutralize KSHV infectivity *in vitro* [21–23], however the association between naturally occuring neutralizing antibodies and development of KS is not clear. KSHV can infect several host cell types, including oral epithelial cells, endothelial cells, B-cells, and other mononuclear cells [24,25]. KSHV entry depends on the interaction of several virally encoded glycoproteins with multiple host cell surface receptors that define tropism

**Competing interests:** I have read the journal's policy and the authors of this manuscript have the following competing interests: ATM, JB, WP and MP have submitted a US provisional application related to the anti-gH/gL mAbs described herein.

and mediate a complicated process of viral internalization and membrane fusion [25,26].

Five KSHV glycoproteins are considered essential for entry: K8.1A/B, gH, gL, and gB. K8.1A and K8.1B are splice variants encoded by the same viral open reading frame (ORF). K8.1A and K8.1B attach to target cells through binding to heparan sulfate proteoglycans [27–29]. K8.1 is also essential for entry into some B cell lines in a heparan-sulfate-independent mechanism, and plays an important role in attachment to certain epithelial cell lines suggesting that additional unknown receptors may be used by K8.1 during viral entry [30,31]. Other studies have demonstrated that K8.1 is not essential for infection of epithelial cells or B cells [32,33], highlighting a lack of consensus on the role of this viral glycoportein in infection.

gH, gL, and gB comprise the core fusion machinery which is conserved across herpesviruses. These proteins are essential for mediating host-viral membrane fusion. gB is a trimeric fusion protein that mediates the merger of the host and viral membranes [34]. In addition to its direct role in fusion, gB participates in viral attachment by making direct interactions with heparan sulfate [35] as well as α3β1, αVβ3, αVβ1, and αVβ5 integrins on the cell surface [36–39].

For most herpesviruses, the fusion activity of gB depends on gH and gL. gH and gL form a complex (gH/gL) that triggers gB following the engagement of one or more host cellular receptors [34]. Cellular receptors bound by KSHV gH/gL include ephrin type A2, A4, A5, and A7 receptors (EphA2, EphA4, EphA5, and EphA7) [40–44]. The molecular details of the gH/gL-EphA2 interaction have been elucidated through X-ray crystallography demonstrating that the interaction is primarily mediated by gL [45,46]. Recombinant virus lacking gH is unable to enter epithelial cells, endothelial cells, and fibroblasts, confirming that it is essential for infection [47], while recombinant virus harboring mutations in gH that abrogate EphA2 binding remain infectious [42], suggesting that other receptors and/or functional sites exist on gH/gL.

A subset of persons with KSHV infection develop plasma neutralizing antibodies that can block viral entry *in vitro* [21–23,48]. Determining the viral surface glycoproteins targeted by neutralizing antibodies would aid in vacccine design. This is challenging, however, because the function of the viral entry machinery is spread across multiple glycoproteins and is further complicated by the diverse tropism of KSHV and a poor understanding of the mechanics of viral entry. Nevertheless, serum depletion studies have identified the gH/gL glycoprotein complex as target of antibodies that neutralize KSHV infection of 293 cells and SLK cells implicating gH/gL as a relevant target for vaccine development [23,49].

Here we utilized serum and PBMC samples from a prospective KSHV transmission cohort in Uganda to characterize gH/gL binding antibodies and KSHV neutralizing activity in the sera. Using antigen-specific single B cell sorting, we isolated 12 monoclonal antibodies (mAbs) that map to 5 distinct epitope regions on gH/gL. Negative stain electron microscopy of mAb/gH/gL complexes defined 3 neutralizing epitopes on KSHV gH/gL, including the EphA2 binding site. These findings define critical sites of vulnerability on KSHV that are relevant for KSHV vaccine design.

## Results

### Production of recombinant gH/gL in 293 cells

We produced a soluble recombinant gH/gL ectodomain in 293 cells by co-transfecting plasmids encoding each protein (Fig 1A). gH/gL was purified by immobilized metal affinity chromatography followed by size exclusion chromatography (SEC). KSHV gH can express without gL [50]. Therefore, the purified protein could be a mix of gH and gH/gL. To assess this, we analyzed fractions from the leading and tailing edge of a gH/gL peak from size exclusion chromatography by SDS-PAGE (Fig 1B). We observed differences in the electrophoretic mobility of gH, likely due to the presence of different glycoforms, and/or to proteolytic clipping, as evidenced by a band at ~35kDa in the later fractions that was not present in the early fraction (Fig 1C). gL was present in comparable amounts in all fractions indicating that it was incorporated uniformly in our gH/gL preparation (Fig 1C). SDS-PAGE analysis of the purified protein revealed the presence of a band corresponding to gH at the expected molecular weight of ~100 kDa as well as two lower molecular weight (LMW) bands, one slightly larger, and one slightly smaller than 25 kDa (Fig 1C and 1D). gH/gL treated with PNGaseF migrated as two distinct species at ~75 and ~17kDa. (Fig 1D), consistent with the presence of glycans on recombinant gH/gL. This result suggests that the two lower molecular weight bands in the untreated sample correspond to two different glycoforms of gL. To confirm this, we excised both LMW bands from an SDS-PAGE gel and verified that they were both gL by mass spectrometry (S1 Fig). To ensure the presence of properly folded gH/gL complexes, we confirmed binding to recombinant EphA2 by biolayer interferometry (BLI) while an irrelevant protein, gp350 from EBV, did not (Fig 1E). Collectively, these analyses demonstrate that the recombinant KSHV gH/gL is glycosylated and capable of binding its cognate receptor.

### Serum reactivity to gH/gL in persons with KSHV infection

To measure serum reactivity to recombinant gH/gL, we selected 23 plasma samples from a prospective cohort of 80 Ugandan households (mothers and up to 4 children) that were followed for up to 1 year with weekly home visits to characterize KSHV shedding dynamics and identify factors associated with KSHV transmission. Oral swabs were collected weekly, and blood samples were collected at enrollment and every 3 months. We evaluated sera from 23 participants, including 9 mothers and 7 children with documented KSHV infection based on detection of KSHV DNA by quantitative PCR in blood or saliva [51] and 7 children without documented KSHV infection. All KSHV PCR⁺ mothers had readily detectable reciprocal endpoint antibody binding titers to gH/gL (Figs 2A and S2). The titers among the KSHV PCR⁺ children were more variable, ranging from just above 1:20 to ~1:5000 (Figs 2A and S2). 4/7 of the KSHV PCR⁻ children were gH/gL seronegative while the remaining three had low reactivity with reciprocal endpoint titers below 1:50, (Figs 2A and S2). Overall, the endpoint titers of the PCR⁻ children were comparable to those measured in participants from a Seattle-area cohort of adults that are presumed to be KSHV⁻ (Fig 2A gray). These gH/gL plasma antibody binding data are largely consistent with the KSHV status as determined by PCR in the Ugandan cohort. We also evaluated the ability of the plasma to neutralize KSHV infection of Vero cells at a single, low dilution. All plasma from the KSHV PCR⁺ mothers could neutralize infectivity by at least 80% (Fig 2B). Three of the plasma samples from KSHV PCR⁺ children (044-C2, 041-C2 and 026-C3) could also reduce infectivity by >80% (Fig 2B). These same 3 samples had gH/gL binding titers above 1:1000 (Fig 2A). None of the presumed KSHV⁻ participants neutralized KSHV infection, and none of the plasma samples from KSHV PCR⁻ children could reduce infectivity by 80% although two, 009-C1 and 039-C1 could reduce infectivity by >50% (Fig 2B). 009-C1 had very weak gH/gL binding titers, while 039-C1 had no gH/gL binding titers (Fig 2A). Weak neutralizing activity observed in these two participants may reflect early KSHV infection that was not detectable in blood or saliva by PCR testing. Overall, the plasma neutralizing activity correlated with gH/gL binding titers in the Ugandan cohort (Fig 2C).

   Collectively, these data indicate that plasma antibodies from the majority of participants with KSHV infection recognize recombinant gH/gL confirming the antigenicity of gH/gL. They further suggest that high titers of gH/gL binding antibodies

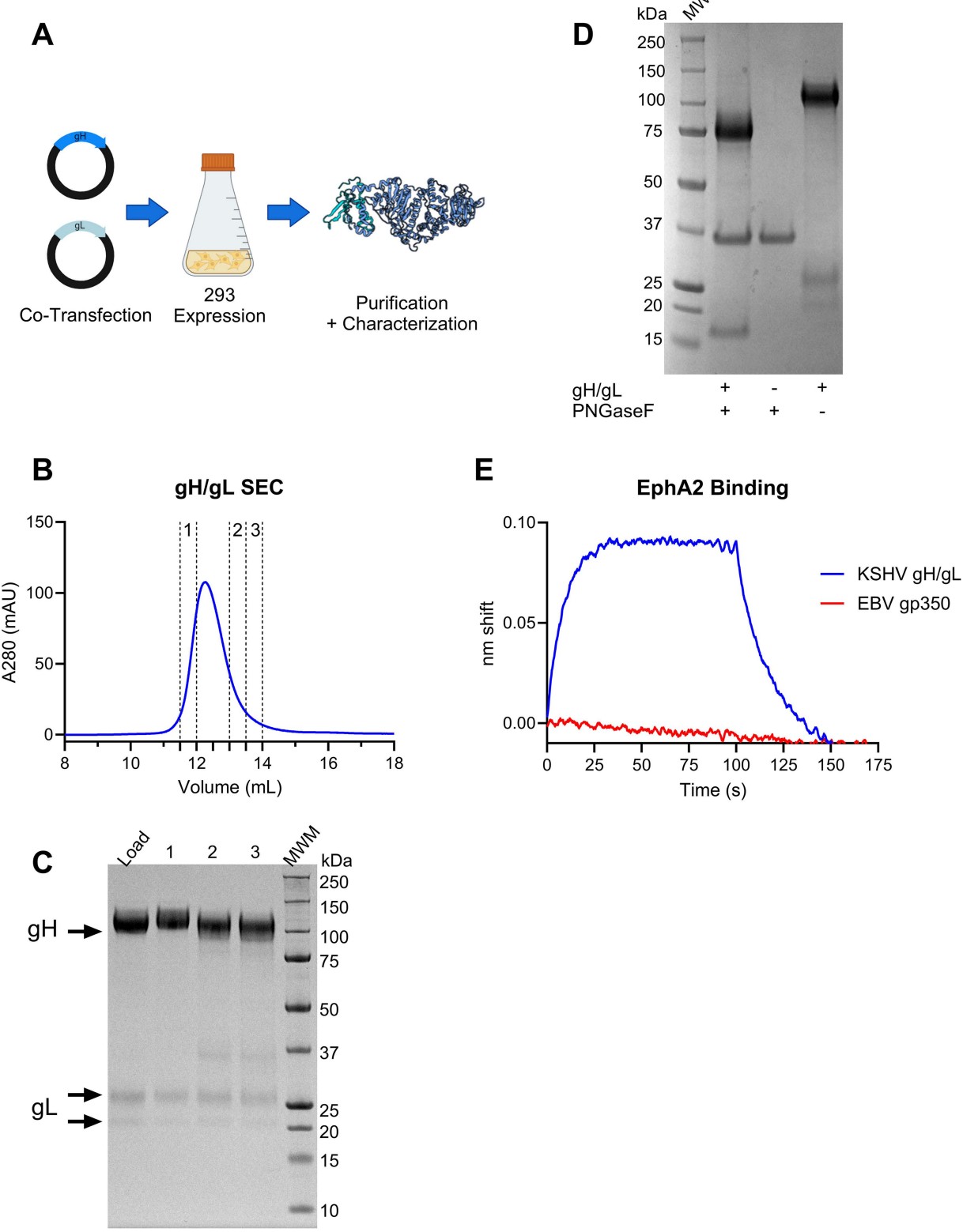

**Fig 1. Expression, purification and characterization of recombinant KSHV gH/gL.** (A) Plasmids encoding the gH ectodomain and gL were co-transfected into 293 cells and recombinant gH/gL was purified from the supernatant. Made with BioRender.com. McGuire, A. (2025). https://BioRender.

com/ks0xtn4. (B) Recombinant gH/gL was subjected to size exclusion chromatography on a BioRad ENrichSEC 650 10 x 300 column. Three fractions were collected as indicated. (C) An aliquot of the gH/gL preparation pre-SEC, as well as equal amounts of the three fractions collected from B were analyzed by reducing SDS-PAGE followed by Coomassie staining. (D) Coomassie stained reducing SDS-PAGE gel of recombinant gH/gL treated, or untreated with PNGaseF as indicated. (E) Binding of recombinant gH/gL or recombinant Epstein-Barr virus gp350 to recombinant ephrin receptor A2 (EphA2) was measured by biolayer interferometry.

are associated with neutralizing activity in epithelial cells, supporting the notion that gH/gL is a major target of neutralizing antibodies [23,49].

## Isolation of gH/gL specific memory B cells

To obtain a more granular understanding of the anti-gH/gL antibody response to KSHV infection, we sought to isolate recombinant monoclonal antibodies (mAbs) encoded by gH/gL-specific memory B cells from KSHV PCR+ participants. To this end, gH/gL was biotinylated, conjugated to both streptavidin-phycoerythrin (PE) and streptavidin-allophycocyanin (APC) and used to stain memory B cells (Fig 3A) from PBMCs. PBMCs from two KSHV+ participants from the KSHV transmission cohort with relatively high gH/gL binding titers (Fig 2A, U035-22-MO and U035-67-MO) were chosen as well as PBMCs from a donor with no known history of KSHV infection, and minimal serum reactivity to gH/gL were evaluated. The dual-labeling strategy readily identified a population of gH/gL-specific B cells in a gH/gL seropositive participant, U035-22-M0 (Fig 3B) but not in the sample from KSHV- participant (Presumed Negative A, Figs 2A-2B and S2) demon-strating that it can specifically label gH/gL-specific B cells (Fig 3B and 3C). gH/gL-positive B cells were single-cell sorted and their antibody variable region heavy and light chain transcripts were recovered by RT-PCR, cloned into expression vectors to facilitate mAb production, and sequenced to determine gene usage and somatic mutation rates [52,53]. Using this approach, we successfully isolated and expressed 12 mAbs. All 12 mAbs bound to gH/gL from KSHV (Fig 3D) but not EBV (Fig 3E) with affinities ranging from sub-micromolar to sub-nanomolar (Figs 3F and S4).

Sequence analysis revealed that 2 mAbs, MLKH1 and MLKH5, isolated from the same participant are clonally related. They are derived from a VH1–69 heavy chain and a lambda 7–43 light chain with similar CDRH3 and CDRL3 junctions (S3). An additional 3 mAbs from this participant (MLKH6, MLKH7 and MLKH9) are derived from a VH4–61 heavy chain paired with a kappa 3–15 light chain and are predicted to have the same D and J gene usage (S3), however their CDRH3 and CDRL3 junctions are dissimilar (S3) indicating that they may be clonally related, or that this gene usage represents a reproducible class of gH/gL-specific antibodies. The remaining mAbs have diverse VH and VL gene usage and CDR3 sequences (S3). The mAbs were somatically mutated, showing ~2–20% divergence from their predicted germline VH and VL genes at the amino acid level (Fig 3G).

## Epitope cluster analysis

To determine whether the isolated mAbs bound overlapping or distinct epitopes, we used BLI to assess whether the mAbs competed each other for binding to gH/gL (Fig 4). gH/gL was immobilized on a biosensor and then incubated with the mAb of interest until saturable binding was achieved, then the biosensor was immersed into a well with a second mAb. Binding inhibition was measured as reduction in binding of the second mAb relative to its binding to gH/gL alone. This cross-competition analysis revealed that the mAbs map to 5 epitope clusters. These include Cluster 1, bound by MLKH1 and MLKH5 (Fig 4, blue). MLKH4, MLKH6, MLKH7, MLKH9, and MLKH11 make up Cluster 2 (Fig 4, red). Cluster 3 includes MLKH12 and MLKH2 (Fig 4, purple). MLKH12 competes with all Cluster 2 mAbs except MLKH11. In contrast, MLKH2 does not compete with any Cluster 2 mAbs. We therefore deduce that Cluster 3 partially overlaps with Cluster 2. Cluster 4 includes MLKH3 and MLKH8 (Fig 4, Teal). Cluster 5 is defined solely by MLKH10 which did not compete with any other mAbs (Fig 4, orange). Among these clusters, mAbs in Cluster 1 are clonal relatives (S1 Table). Similarly, 3 of the 6 mAbs in Cluster 2, MLKH6, MLKH7, and MLKH9 share genetic characteristics (S1 Table).

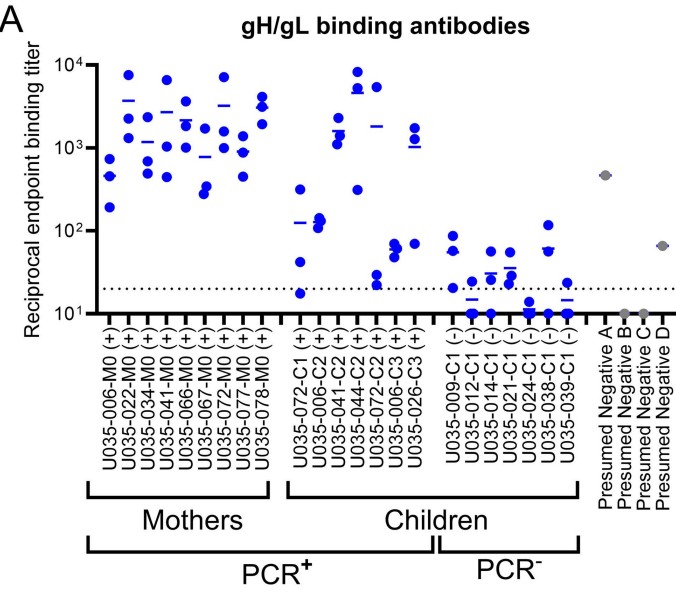

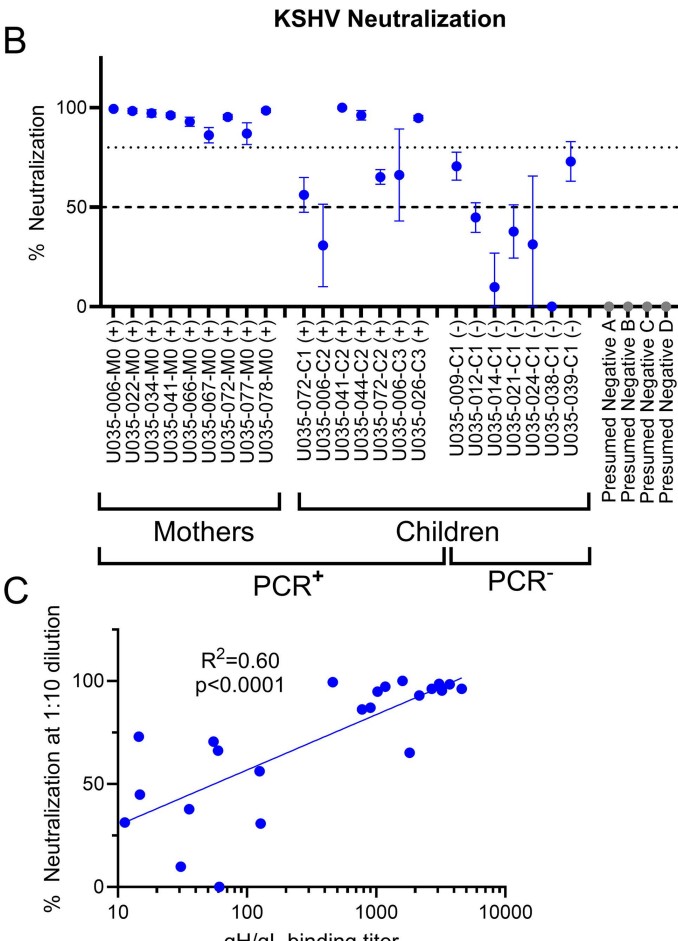

**Fig 2. gH/gL binding and neutralizing activity in a prospective KSHV-infection cohort.** (A) Plasma collected from a prospective cohort of mothers and their children was evaluated for binding to gH/gL by ELISA. Reciprocal endpoint titers are shown, each dot represents an independent measurement

done in duplicate and the bars represent the means. The dashed line indicates the lowest serum dilution tested. Serum from healthy Seattle-area adults that are presumed to be KSHV-negative are included as controls (gray dots). (B) Serum samples from A were evaluated for their ability to neutralize KSHV infection of Vero cells at a 1:10 dilution. Dashed and dotted lines indicate 50% and 80% neutralization respectively. Samples denoted with (+) tested positive for KSHV DNA in contemporaneously collected oral swabs or blood by PCR. "M" indicates that the sample was collected from a mother and C1, C2, and C3 indicate that the sample was collected from a child. The numbers indicate the family household. (C) Correlation analysis of neutralizing activity vs gH/gL binding titer. The mean neutralizing activity and binding titers were plotted on the Y and X axis and analyzed by linear regression.

## Neutralization assays

We evaluated the ability of the mAbs to neutralize KSHV infection in Vero cells. For this analysis, we used the recombinant KSHV.219 strain which expresses a GFP reporter [54]. We initially screened the mAbs for neutralizing activity at a concentration of 100 µg/mL. MLKH1, MLKH5, MLKH6, MLKH7, MLKH11, MLKH12 and MLKH3 reduced infectivity by >80% (Fig 5A) and were subsequently titrated to measure their relative potencies.

The Cluster 1 mAbs MLKH1 and MLKH5 were the most potent, and neutralized KSHV in Vero cells with mean half-maximal inhibitory concentrations ($IC_{50}$) of 0.03 µg/mL (Figs 5B, 5I and S4) and 0.02 µg/mL, respectively (Figs 5C ,5I and S4). Among the Cluster 2 mAbs, three were neutralizing. Of the three genetically similar Cluster 2 mAbs, MLKH6 and MLKH7 had mean $IC_{50}$ values of 0.06 µg/mL (Figs 5D, 5I and S4) and 1.0 µg/mL (Figs 5E, 5I and S4), while MLKH9, was non-neutralizing (Fig 5A). MLKH9 binds gH/gL with ~1000-fold lower affinity than MLKH6 and MLKH7 (Fig 3F), which may explain its lack of neutralizing activity. The genetically distinct Cluster 2 mAb MLKH11 neutralized with a potency of 0.24 µg/mL (Figs 5F, 5I and S4) while the remaining Cluster 2 mAb MLKH4 was non-neutralizing (Fig 5A). The cluster 3 mAb MLKH12 neutralized with a potency of 1.65 µg/mL (Figs 5G, 5I and S4). Despite overlapping with MLKH12, the MLKH2 mAb was non-neutralizing. Similarly, the Cluster 5 mAb, MLKH10 was non-neutralizing (Fig 5A). The potency of the Cluster 4 mAb MLKH3 was 0.08 µg/ml (Figs 5H, 5I and S4). The other Cluster 4 mAb, MLKH8 was non-neutralizing (Fig 5A). The lack of neutralizing activity by MLKH8 may be related to the fact that it binds to gH/gL with ~1000-fold lower affinity than MLKH3 (Fig 3F).

Given that Vero cells are of simian origin, we next assessed the ability of the mAbs to inhibit membrane fusion using a virus free syncytia assay with a human epithelial target cell (293T) [43]. CHOK-1 cells were transfected with plasmids encoding KSHV gH, KSHV gL, and EBV gB, as well as a luciferase reporter gene under the control of T7 polymerase. EBV gB was used here since KSHV gB is unable to mediate fusion in this assay, likely due to a low pH requirement for fusion [43,55]. Twenty-four hours later, the cells were overlaid on 293 cells stably expressing T7 polymerase with or without anti-gH/gL or control mAbs (Fig 5J). The addition of a control mAb, AMMO1 had no effect on syncytia formation. In contrast, preincubation with all of the mAbs except MLKH4 and MLKH8 reduced syncytia formation by greater than 50% (Fig 5J). Interestingly, MLKH9, MLKH2 and MLKH10, had weak to absent neutralizing activity (Fig 5A) but antagonized syncytia formation.

Lastly, we evaluated the ability of the mAbs to neutralize KSHV infection of B cells using the RPMI8226 line [56] at a mAb concentration of 100 µg/mL. None of the mAbs were able to substantially reduce infectivity (i.e., >50% reduction) in this assay (Fig 5K). Similarly, the 4A4 mAb, which can neutralize infection of another B cell line MC116 [31] was non-neutralizing (Fig 5K). Collectively these data indicate that with the exception of MLKH 10 (Cluster 5), at least one mAb from each cluster can neutralize KSHV infection of epithelial cells, but none can neutralize KSHV infection of the RPMI8226 B cell line.

## Cluster 1 neutralizing mAbs inhibit the gH/gL-EphA2 interaction

EphA2 is an entry receptor for KSHV that binds primarily through residues of gL in the gH/gL complex [44–46]. Therefore, we assessed whether any of the mAbs could inhibit the gH/gL EphA2 interaction using BLI. EphA2 binding to gH/gL was uninhibited by the presence of an isotype control mAb, AMMO1 [57] (Fig 6A). In contrast, EphA2 was unable to bind gH/

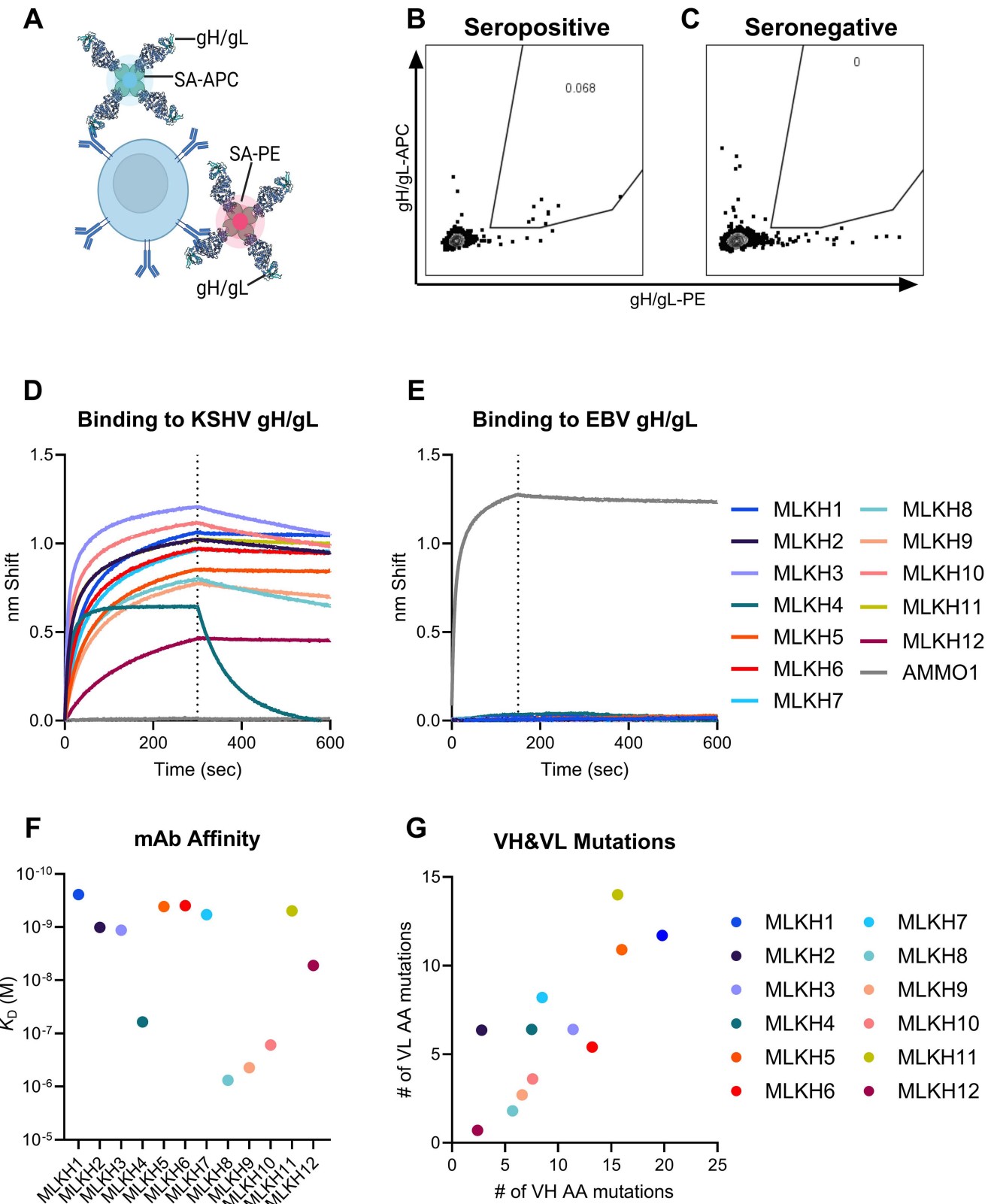

**Fig 3. Isolation of gH/gL-specific mAbs.** (A) Schematic of memory B cell staining with phycoerythrin (PE) and allophycocyanin (APC) conjugated gH/gL tetramers. Created with BioRender.com. McGuire, A. (2025) https://BioRender.com/bh8ls7o. (B) Staining of memory B cells (live, CD3⁻, CD14⁻,

CD19⁺, IgD⁻ IgG⁺) from a KSHV gH/gL seropositive donor, U35-022-MO (from Fig 2), with APC- and PE-labeled gH/gL. (C) gH/gL staining of memory B cells from a presumed KSHV negative donor (Presumed Negative A, from Fig 2). (D) Biolayer interferometry traces of gH/gL mAbs isolated from single-cell sorted gH/gL-labeled B cells binding to a 500 nM solution of KSHV gH/gL. (E) Biolayer interferometry traces of gH/gL mAbs in D binding to a 500 nM solution of EBV gH/gL. The anti-EBV gH/gL mAb AMMO1 was used as a control, and the dashed line demarcates the association and dissociation steps in D and E. (F) Affinity of the indicated mAbs to serially diluted KSHV gH/gL were measured by biolayer interferometry. (G) The number of amino acid mutations found in the antibody variable heavy (VH) and variable light chain (VL) chain genes of the indicated mAbs.

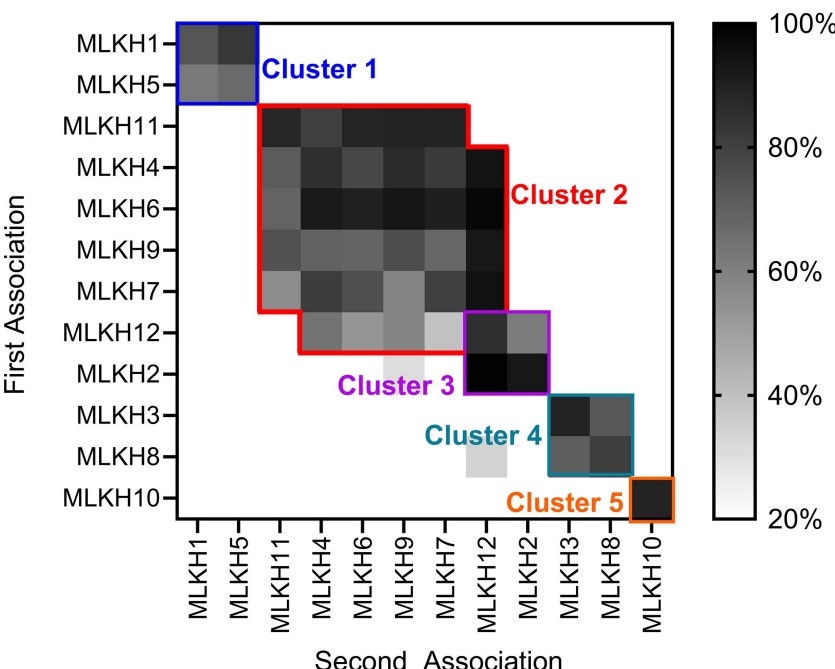

**Fig 4. Epitope binning of gH/gL-specific mAbs.** Heat map depicting the competition of KSHV mAbs binding to gH/gL as determined by BLI. gHgL was biotinylated and immobilized on a streptavidin biosensor and then immersed in buffer containing the indicated mAb until saturable binding was achieved (y-axis), the biosensor was then immersed in buffer containing the second indicated mAb (x-axis). The percent of binding inhibition is shown as a percentage of residual binding in the presence of the second mAb relative to the binding of the first mAb to gH/gL alone (scale at right). Colored outlines represent epitope clusters as determined by competition BLIs.

gL when preincubated with an excess of either MLKH1 or MLKH5 (Fig 6A). When gH/gL was preincubated with any of the other gH/gL mAbs, enhanced binding to EphA2 was observed (Fig 6A). This binding-enhancement is consistent with binding to epitopes outside of the EphA2 binding footprint resulting in higher avidity binding through immune complex formation.

To confirm that these mAbs bind to the EphA2 binding site, we used negative-stain electron microscopy (nsEM) to visualize the MLKH1/gH/gL and MLKH5/gH/gL complexes. The 2D class averages revealed the gH/gL 4 domain structure with the antigen binding fragment (Fab) bound to one end (Fig 6B and 6C). The crystal structure of the EphA2/gH/gL complex is shown for comparison which supports that the mode of interaction between MLKH1, MLKH5 and EphA2 with gH/gL are similar (Fig 6B-6D). Collectively these binding and structural analyses are consistent with MLKH1 and MLKH5 binding to an epitope that overlaps with the EphA2 binding site, providing a plausible mechanism of neutralization.

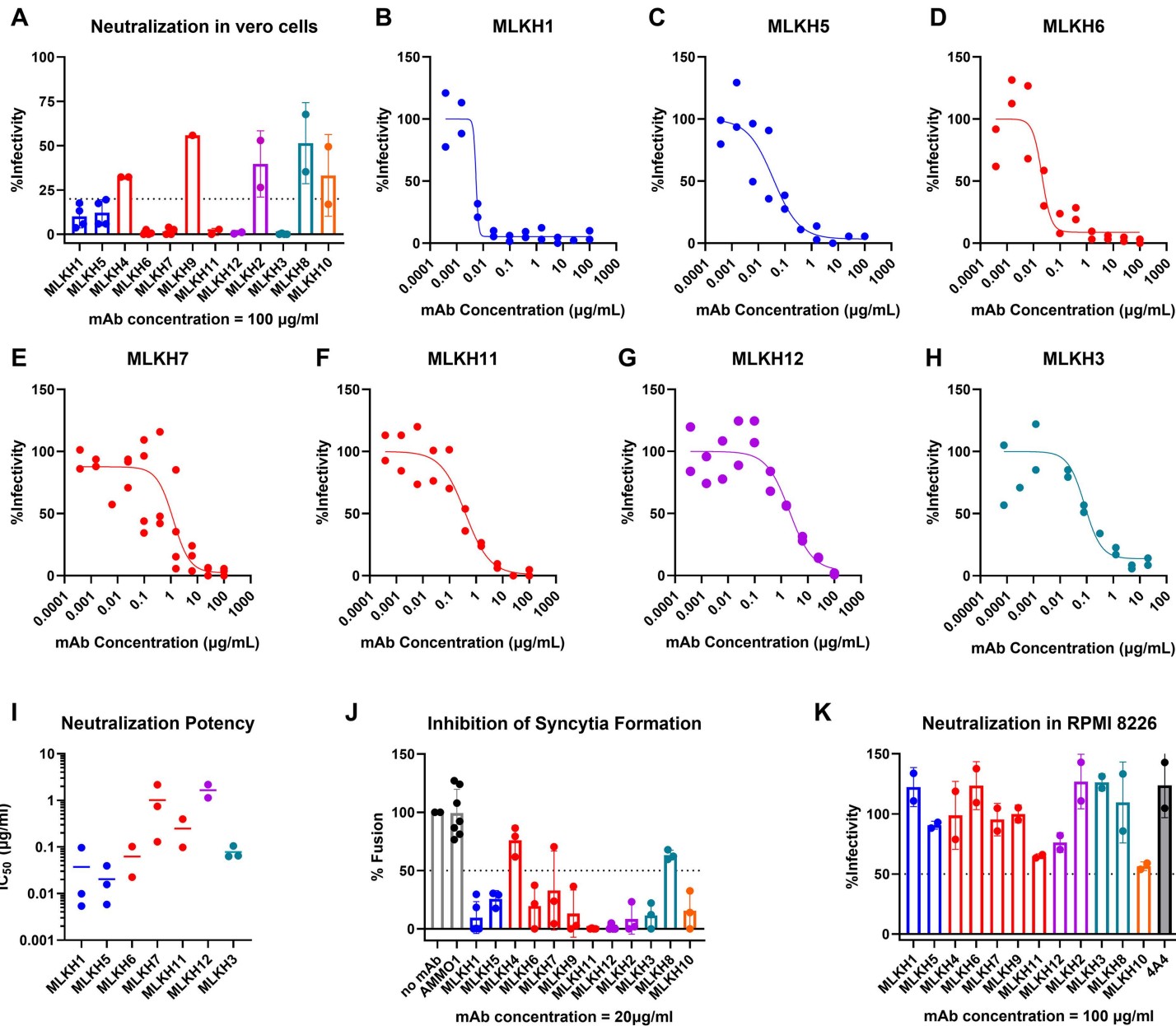

**Fig 5. Neutralization by gH/gL-specific mAbs.** (A) Recombinant mAbs were evaluated for their ability to neutralize KSHV infection of Vero cells at a single concentration of 100 μg/mL cells as indicated. Coloring of mAbs represent epitope cluster determined in Fig 4. (B-H) mAbs with neutralizing activity from A were serially diluted and tested for neutralization against KSHV infection of Vero cells as indicated. Each data point represents a technical replicate from one representative experiment. (I) The half-maximal inhibitory concentration ($IC_{50}$) of the indicated mAbs was calculated from the curves in B-H. Each dot represents the $IC_{50}$ calculated from an independent experiment carried out in duplicate, the bar represents the mean. (J) The indicated recombinant mAbs were evaluated for their ability to inhibit syncytia formation in a virus-free fusion assay as indicated. Each dot represents the luciferase activity in the presence of each mAb relative to the fusion measured in the absence of mAb. Each dot represents the mean of an independent experiment measured in triplicate. Bars represent the mean and error bars represent standard deviation. (K) The indicated recombinant mAbs were evaluated for their ability to neutralize KSHV infection of RPMI8226 cells as indicated.

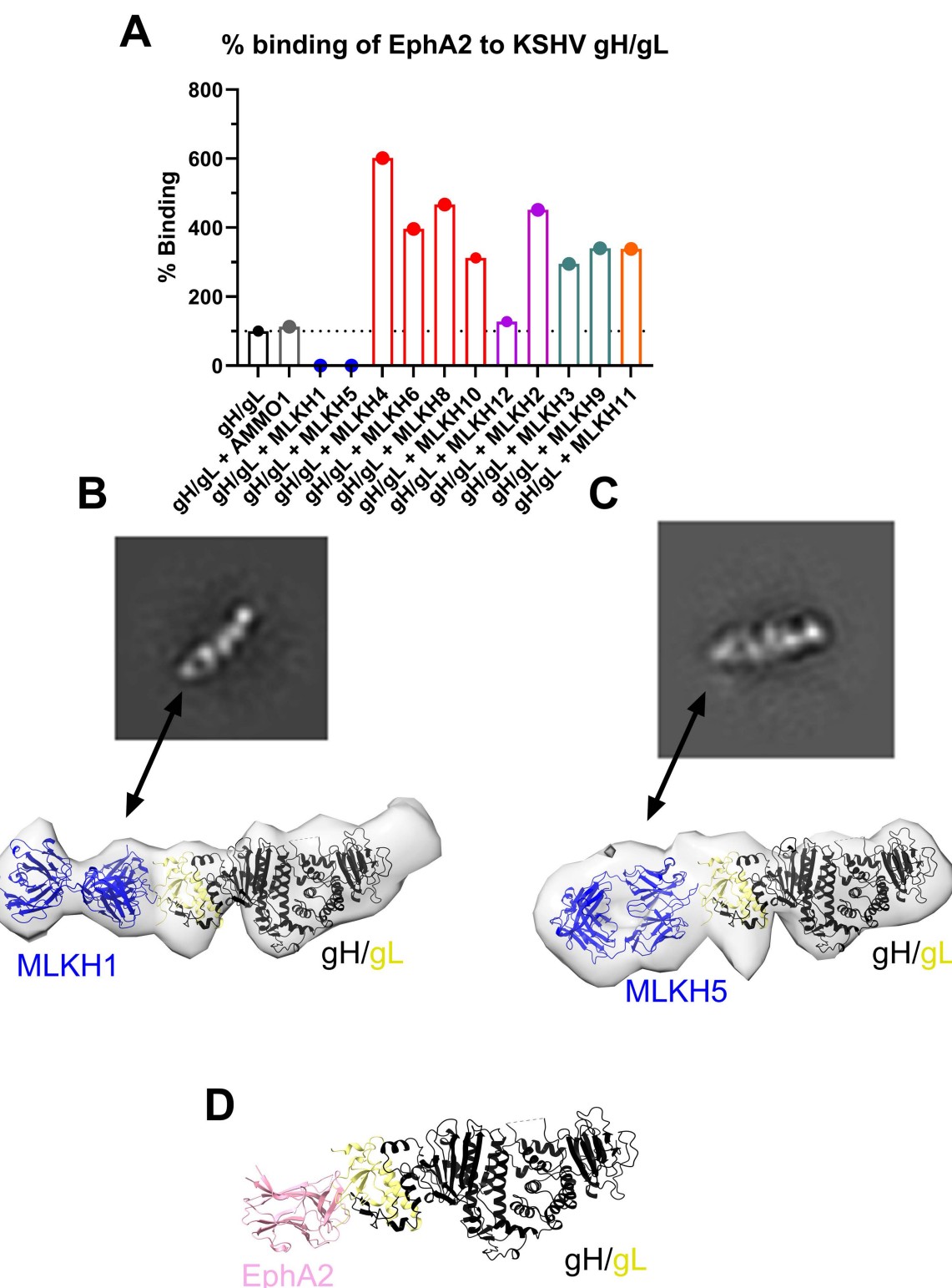

**Fig 6. Potent neutralizing mAbs target the EphA2 binding site.** (A) gH/gL was incubated with, or without the indicated mAbs and the binding to immobilized EphA2 was measured by biolayer interferometry. The dashed line represents 100% binding set to the gH/gL alone control. (B-C) MLKH1/gH/gL (B) and MLKH5/gH/gL (C) complexes were purified and visualized by negative stain EM (nsEM). A representative 2D class average (top) and 3D

reconstruction (bottom) are shown for each complex. The ribbon structure of gH/gL (PDB entry 7CZF, gH in black, gL in yellow) with each Fab (modeled using Alphafold) was fitted into the nsEM maps. (D) The crystal structure of the EphA2/gH/gL complex (PDB entry 7CZE) is shown for comparison.

## Mapping of additional neutralizing epitopes by nsEM

We sought to delineate the epitopes of other neutralizing mAbs on gH/gL. To this end, we carried out nsEM of gH/gL complexed with the MLKH3, MLKH6 and MLKH12 Fabs. For this analysis, we included the non-competing MLKH5 and MLKH10 Fabs to facilitate the orientation of the complexes. Three-dimensional reconstructions of the gH/gL/MLKH5/MLKH10/MLKH3 complex revealed that MLKH10 and MLKH3 bind to D-II and D-III/D-IV of gH/gL, respectively (Fig 7A and 7D). The gH/gL/MLKH5/MLKH10/MLKH6 complex revealed that MLKH6 also binds to D-IV, but to a different epitope on the distal face of D-IV (Fig 7B). Consistent with their competition for binding to gH/gL (Fig 4), MLKH12 binds to a similar epitope as MLKH6 on D-IV, but with a different angle of approach (Fig 7C and 7D). These data indicate that D-IV harbors additional neutralizing epitopes defined by MLKH3, MLKH6 and MLKH12.

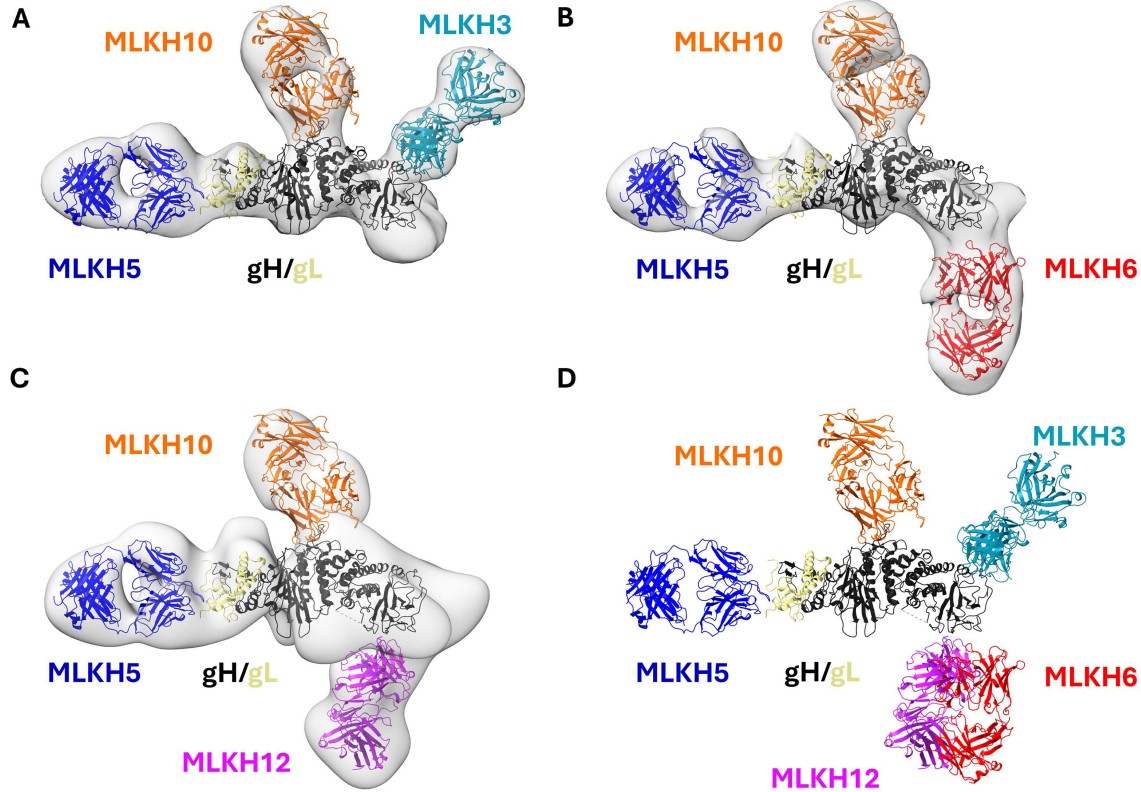

**Fig 7. Neutralizing anti-KSHV gH/gL mAbs target multiple epitopes on gH/gL.** (A-C) gH/gL was incubated with MLKH5, MLK10 and MLK3 Fabs (A), MLKH5, MLKH10 and MLKH6 (B) or MLKH5, MLKH10 and MLKH12 (C), and complexes were visualized by nsEM. A representative three-dimensional reconstruction is shown for each complex. A comparison of the different complex is also shown (D). The ribbon structure of gH/gL (PDB entry 7CZF, gH in black, gL in yellow) along with the different Fabs models predicted with Alphafold (MLKH3 in teal, MLKH5 in blue, MLKH6 in red, and MLKH10 in orange and MLKH12 in purple) were fitted into the nsEM maps.

## Cryo-EM structure of MLKH3 and MLKH10 antibodies

Given that the gH/gL/MLKH5/MLKH10/MLKH3 complex harbored two of the most potent neutralizing mAbs we identified, we sought to obtain higher resolution structures using cryogenic electron microscopy (cryo-EM). While we could see the three Fabs bound (S3 Fig), we were unable to achieve high resolution around the MLKH5 Fab and epitope (S3F-S3G Fig). We also observed a class of particles that appeared to lack gL (S3A Fig) indicating that the co-transfection of gH and gL on separate plasmids did indeed result in a mixed pool of gH and gH/gL that we were unable to resolve by size exclusion chromatography (Fig 1B and 1C). Nevertheless, we were able to obtain structures of MLKH3 and MLKH10 in complex with gH/gL to 3.51 Å (Figs 8A, 8B, and S3 and S3 Table) with local resolution at the interface of 2.8 Å and 2.7 Å for MLKH3 and MLKH10, respectively (S3F and S3G Fig). MLKH3 binds to D-III and D-IV of gH, protruding perpendicularly and interacts exclusively through its heavy chain. The MLKH3–gH interface has a total buried surface area (BSA) of ~646 Å² on gH (Fig 8C) and ~691 Å² on MLKH3 (Fig 8E), with all heavy-chain CDRs involved in the interaction (Figs 8E and S4A–S4C). MLKH10 binds to D-II of gH and interacts with one residue in gL (Fig 8D). Both the heavy and light chains participate in the binding, with a total BSA of approximately ~1171 Å² on gH (Fig 8D), ~701 Å² on MLKH10 heavy-chain (Fig 8F) and ~477 Å² on MLKH10 light chain (Fig 8G). All six CDRs are involved in the MLKH10 interaction (Figs 8F–8G and S4D–S4I).

## Discussion

Like all herpesviruses, KSHV infection is critically dependent on the core fusion machinery comprised of gH, gL and gB. Among these, gH is essential for infection [47]. This suggests that components of the core fusion machinery, and gH/gL in particular, are reasonable candidates for vaccine development [15] or targets for therapeutic interventions. Indeed, the gH/gL glycoprotein complex is a major target of neutralizing antibodies elicited by natural infection [23,49]. However, the precise epitopes targeted by gH/gL neutralizing mAbs are poorly characterized and a better understanding of them could facilitate vaccine design. In this study, we isolated several anti-gH/gL mAbs from persons with KSHV infection, measured their neutralizing activities and used biochemical and structural analyses to define sites of vulnerability on gH/gL.

gH/gL interacts with EphA2 and other Eph family receptors [41–44]. Most of the gH/gL/EphA2 contacts are mediated by gL encoded resides on D-I [45,46]. Two of the most potent neutralizing mAbs isolated here block EphA2 binding to gH/gL by binding to an epitope in Cluster 1 that directly overlaps with the EphA2 binding site. This highlights the importance of this interaction for KSHV infection and confirms that the EphA2 binding site is a target of neutralizing antibodies [58]. Consequently, the EphA2 binding site is likely relevant to vaccine design and could be a site of vulnerability targeted by immunotherapy approaches or small molecule inhibitors for the treatment of KSHV and associated diseases.

KSHV can also infect cells in an EphA2 independent manner [42]. Mice immunized with recombinant gH/gL developed serum antibodies that could neutralize KSHV harboring mutations that fail to efficiently incorporate gL into gH/gL complexes abrogating EphA2 binding [42,58]. This intimates that neutralizing epitopes outside the gH/gL EphA2 binding site exist on gH/gL. Here we identify 5 neutralizing mAbs that bind such epitopes. Cryo-EM elucidated that one of these mAbs, MLKH3, binds to an epitope bridging D-III and D-IV and competes for binding with MLKH8, which is non-neutralizing. All other neutralizing mAbs map to a partially overlapping epitope cluster (Cluster 2). nsEM reveals that the Cluster 2 mAb, MLKH6 and the Cluster 3 mAb MLKH12 bind to the C-terminus of D-IV on gH/gL on the opposite face of the MLKH3 epitope. Within Clusters 2, 3 and 4, there are both neutralizing and non-neutralizing antibodies with varying binding affinities. High-resolution structures of neutralizing, and non-neutralizing mAbs in complex with gH/gL will help to discriminate whether the differences in neutralizing activity between competing mAbs is related to affinity, fine epitope specificities, or both.

Given the limited understanding of the mechanisms by which gH/gL participates in KSHV entry, we are unable to assign a mechanism of neutralizing activity to the non-EphA2 binding site mAbs we isolated. Nevertheless, these mAbs define

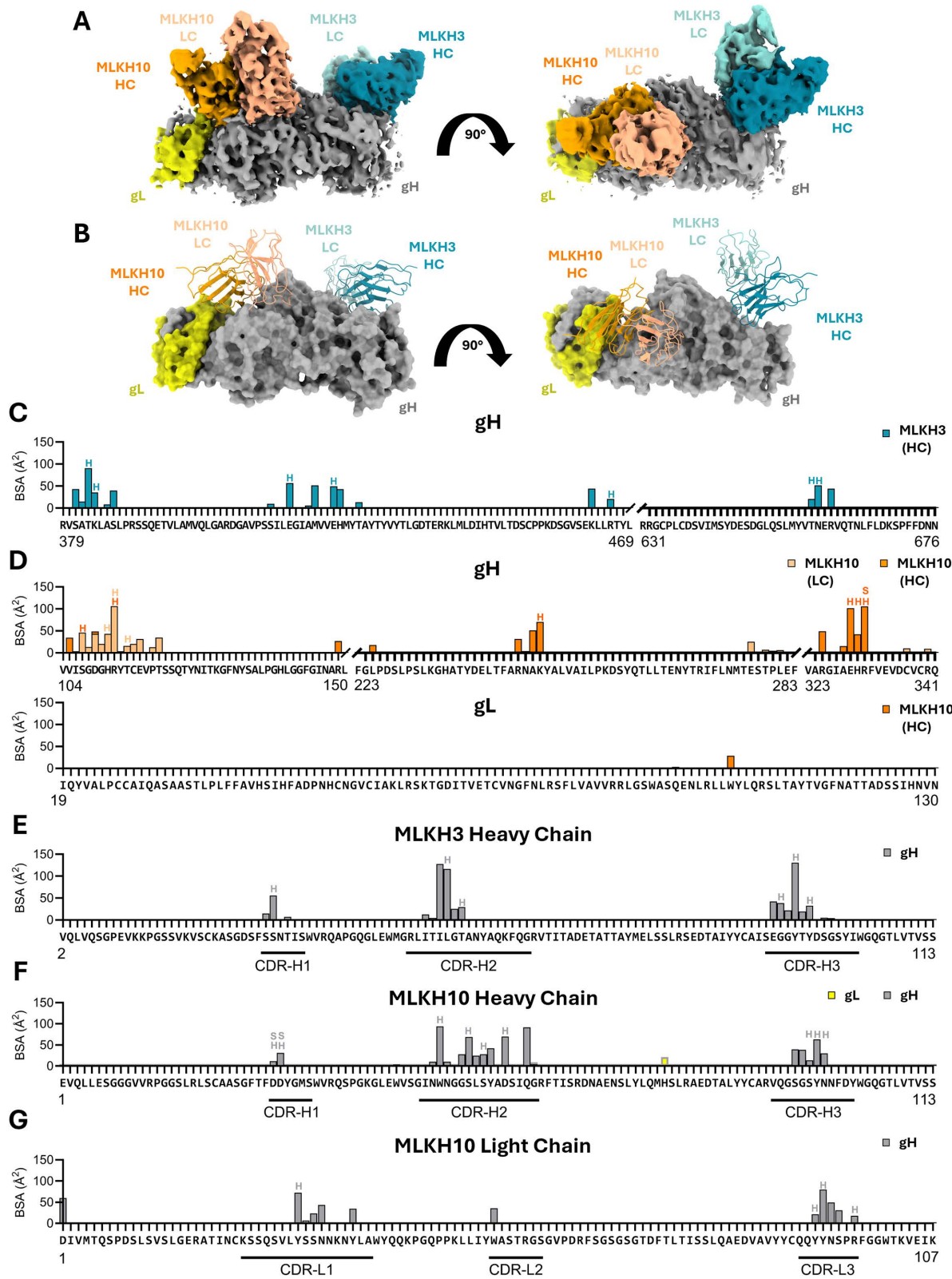

**Fig 8. Cryo-EM structure of MLKH3 and MLKH10 Fab in complex with KSHV gHgL.** (A) Left, cryo-EM map is shown with MLKH10 Fab (VH shown in orange, VL in salmon), MLKH3 Fab (VH shown in teal, VL in light teal), and gHgL in shades of gray and yellow, respectively. Right, 90° view rotation.

MLKH5 Fab is not shown in the map. (B) Structure of the complex with the Fabs shown in ribbons and gHgL as a surface in the left panel; in the right panel, the same complex is shown rotated by 90°, (C–G) BSA plots for the residues of each protein involved in the complex. Hydrogen bonds are marked with an H on top of the bar, and salt bridges are marked with an S. (C) BSA plot for gH and gL residues involved with MLKH10 interaction; (D) BSA plot for gH residues involved with MLKH3 interaction; (E) BSA plot for the MLKH3 heavy chain residues only, as the MLKH3 light chain makes no contact with gH; (F) BSA plot for the residues of the MLKH10 heavy chain; (G) BSA plot for the residues of the MLKH10 light chain.

sites that are critical for gH/gL function and can serve as tools to map vaccine-elicited serum responses and may be useful in deciphering the nature of interactions with alternative receptors, understanding how gH/gL might interact with gB and/or undergo conformational changes necessary for viral entry.

Three mAbs that target an epitope outside the EphA2 binding site, MLKH6, MLKH7 and MLKH9, all share the same VH and VL gene usage and map to the same epitope. MLKH6 and MLKH7 are neutralizing while MLKH9, which has much lower affinity is not. The fact that these mAbs were isolated from the same individual raises the prospect that they could be clonally related. Alternatively, they may define a reproducible class of antibodies that is pre-disposed to bind to gH/gL. Isolating more mAbs with a VH4–61/VK 3–15 gene signature from additional KSHV+ participants would support this notion and indicate a ubiquitous, genetically defined pathway to generate neutralizing antibodies that could be targeted through vaccination. Indeed, reproducible classes of antibodies have been described for other pathogens including HIV-1 [59], SARS-CoV2 [60], RSV [61] and influenza [62].

The varying ability of KSHV to infect different cell lines poses a challenge to our understanding of the entry pathways and to vaccine development. In the present study, we observed that despite antagonizing syncytia formation, MLKH9, MLKH2 and MLKH10 failed to neutralize KSHV infection of Vero cells. This discrepancy may be due to differences in the assays, including receptor expression on target cells (i.e., Vero vs 293), the expression level of gH/gL on transfected cells vs virions, the presence of additional viral proteins in the case of virions, or differences in the ability of gH/gL to activate gB from KSHV vs EBV, including pH sensitivity in the case of KSHV [55].

We also observed that none of the mAbs isolated were able to neutralize KSHV infection of the RPMI8226 B cell line. This suggests that neither the EphA2 binding site, nor the other epitopes defined by the mAbs we identified are functionally critical for KSHV infection of RPMI8226 cells. Antibodies that bind other epitopes on gH/gL or other KSHV glycoproteins might be more important in this regard. We also note that the 4A4 mAb, which has been reported to neutralize infection of the MC116 B cell line in vitro [31] but not human B cells *in vivo,* [33] also failed to neutralize KSHV infection in the RPMI8226 B cell line. Given these discrepancies, further studies to evaluate the neutralizing activity of the gH/gL mAbs against KSHV infection of multiple B cell lines *in vitro* as well as in mice harboring susceptible human B cells are warranted [33,63,64]. Similarly, it will be important to evaluate the ability of the gH/gL mAbs described herein to neutralize KSHV infection of endothelial cells, which are the primary cell type that undergoes malignant transformation when infected with KSHV and leads to KS. Because infection with KSHV is an essential feature of KS tumor spindle cells [65,66], neutralizing mAbs such as the ones described herein that could prevent cell entry may have utility as novel therapeutics for the treatment or prevention of KS.

Herein, we also show that serum gH/gL binding titers generally correlate well with PCR confirmed KSHV infection, which has implications for diagnosis. The K8.1 glycoprotein is thought to mediate attachment to B cells, is unique to KSHV, and has traditionally been used as the basis for serologic determination of KSHV infection/exposure [16,67–73]. The lack of cross reactivity of the mAbs with gH/gL from the other gamma herpesvirus, EBV, suggests that recombinant gH/gL alone or in combination with K8.1 or other viral antigens may have diagnostic value in determining KSHV infection status.

We detected serum neutralizing activity in nearly all mothers with PCR confirmed KSHV infection. This finding is at odds with studies that found that only ~10–20% of asymptomatic KSHV+ donors had KSHV neutralizing titers [21,23], but in agreement with another study that found that neutralizing titers were higher in KSHV asymptomatic KSHV+ participants

as compared to participants with KS [22] or participants in remission from KS [23]. We note that we tested serum at a low dilution and used Vero target cells while the aforementioned studies used 293 cells. Thus, differences in the study populations, assays and serum dilutions tested could account for a lack of concordance across these studies.

In sum, we confirm that gH/gL is a relevant target of KSHV neutralizing antibodies in the serum of individuals with KSHV infection. The structural and functional characterization of gH/gL-specific mAbs from persons with KSHV infection confirms that the EphA2 binding site is important for infection of epithelial cells while additional mAbs define novel sites of vulnerability that are relevant to vaccine design. Moreover, these mAbs can serve as valuable research tools and can form the basis of KSHV-specific therapeutics.

## Methods

### Ethics statement

We tested blood samples from participants enrolled in a prospective cohort of household units (mothers and up to 4 children) who were followed for 1 year with weekly home study visits to characterize KSHV shedding dynamics and identify factors associated with KSHV transmission. Oral swabs were obtained every week to measure KSHV shedding by quantitative RT-PCR. Blood (plasma and PBMC) is obtained every 3 months to measure KSHV copy number and KSHV serology. The cohort enrolled and followed 80 mothers and 174 children in Kampala, Uganda between May 2021 and November 2024. Approval for the study was granted from the Makerere University School of Medicine Research and Ethics Committee (2020–109), The Fred Hutchinson Cancer Research Center Institutional Review Board (FHIRB0010222), and the Uganda National Council for Science and Technology (HS735ES). All participants provided documentation of informed consent or assent. Written parental consent was obtained for all children 17 years or younger; in addition, documentation of assent was obtained for all children 8–17 years of age.

PBMC and sera from the KSHV- control participant were collected from adults without HIV who were recruited at the Seattle HIV Vaccine Trials Unit (Seattle, Washington, USA) as part of the study "Establishing Immunologic Assays for Determining HIV-1 Prevention and Control", also referred to as Seattle Assay Control (SAC) Cohort. All participants signed informed consent, and the Fred Hutchinson Cancer Center (Seattle, Washington, USA) Institutional Review Board approved the SAC protocol (FHIRB0005567) prior to study initiation. Donor was selected randomly and no considerations were made for age or sex.

### Cell lines

HEK293-EBNA1-6E (RRID:CVCL_HF20) cells were cultured in Freestyle 293 expression medium (Thermo Fisher, Cat. # 12338026) and maintained at 37°C and 5% $CO_2$ with gentle shaking at 130 rpm. CHO-K1 (RRID:CVCL_0214) cells were maintained in Ham's F-12 + 10% FBS, 2 mM L-glutamine, 100 U/ml penicillin, and 100 µg/ml streptomycin (cF-12). 293-T7 cells (human female) were maintained in DMEM containing 10% FBS, 2 mM L-glutamine, 100 U/ml penicillin, 100 µg/ml streptomycin (cDMEM) and 100 µg/ml Zeocin (Thermo Fisher, cat# R25001).

RPMI8226 (RRID:CVCL_0014) cells were maintained in RPMI media (Gibco, cat#11875–093) supplemented with 10% Fetal Bovine Serum (FBS) (Cytiva, cat#SH30071.03), penicillin (100U/mL), and streptomycin (100µg/mL) (Gibco, cat#15140–122). Cells were maintained in suspension at 37°C and 5% $CO_2.$

Vero (RRID:CVCL_0059) cells were cultured in cDMEM 37°C and 5% $CO_2.$

### Plasmids

DNA corresponding to amino acids 22–698 of KSHV gH (NCBI: YP_001129375.1) preceded by a TPA leader sequence (MDAMKRGLCCVLLLCGAVFVSPSAS) and followed by a GSGSG linker, a hexa-His tag and an Avi tag were codon optimized, synthesized by Twist biosciences and cloned into pTwist-CMV-OriP to create pTwist-CMV-OriP-KSHV-gH. DNA corresponding to amino acids 21–167 of KSHV gL (NCBI: YP_001129399.1) preceded by a TPA leader sequence

were codon optimized, synthesized by Twist biosciences and cloned into pTwist-CMV-OriP to create pTwist-CMV-OriP-KSHV-gL. Plasmids encoding recombinant antibody heavy and light chains were produced by cloning antibody variable heavy and light chains isolated from sorted B cells (see below) into PCR amplified IgG, IgK or IgL vectors as previously described [53].

## Recombinant protein expression

Plasmids encoding KSHV proteins, or antibody heavy and light chains were transfected into 293-6E cells at a density of $10^6$ cells/ml in Freestyle 293 media using PEI Max (Polysciences Cat# 24765) according to the manufacturer's instructions. Expression was carried out in Freestyle 293 media for 6 days after which cells and cellular debris were removed by centrifugation at $4,000 \times g$ followed by filtration through a 0.22 µm filter. Clarified cell supernatant containing KSHV gH/gL was passed over Ni-NTA resin (GoldBio Cat#H-350–100) pre-equilibrated with Ni-NTA binding buffer (0.5 M NaCl, 10 mM Tris,10 mM imidazole, pH 7.1), followed by extensive washing with Ni-NTA binding buffer, and then eluted with 500 mM imidazole, 0.5 M NaCl, 10 mM Tris, pH 8.0 (Ni-NTA elution buffer). gH/gL was then concentrated and further purified by size exclusion chromatography (SEC) using a 16/600 Superdex 200pg column (Cytiva Lifesciences) equilibrated in PBS. Proteins were flash frozen and stored at -20ºC until use.

Clarified cell supernatant containing recombinant antibodies was passed over Protein A Agarose (GoldBio Cat# P-400–5), followed by extensive washing with PBS, and then eluted with 1 ml of Pierce IgG Elution Buffer (Cat# 21004) into 0.1 ml of 1M Tris HCl, pH 8.0. Purified antibodies were then dialyzed overnight into PBS, flash frozen and stored at -20ºC until use.

To produce Fab fragments, IgG eluted from Protein A agarose was further purified by size exclusion chromatography using a 16/600 Superdex 200pg column (Cytiva Lifesciences) equilibrated in 150 mM NaCl and 5 mM HEPES pH 7.5. Purified recombinant IgG was mixed with LysC (NEB Cat# P8109S) at a ratio of 1µg LysC per 10mg of IgG and incubated at 37°C overnight with Oscillation. The cleaved product was incubated with 1mL of Protein A agarose per 10mg of initial IgG at room temperature for 1 hour to bind any remaining IgG and Fc fragment. The purified Fab was further purified by Size Exclusion Chromatography a 16/600 Superdex 200pg column.

## Mass spectrometry

Purified, recombinant gH/gL was separated by SDS-PAGE, stained with simply blue safe stain and visible bands were excised. Gel slices were washed with water, 50% acetonitrile/50% water, acetonitrile, ammonium bicarbonate (100 mM), followed by 50% acetonitrile/50% ammonium bicarbonate (100 mM). The solution was removed and the gel slices were dried in a speed vac. The gel slices were reduced with dithiothreitol (10 mM in 100 mM ammonium bicarbonate) at 56 ºC for 45 min. The solution was removed and discarded. The gel slices were alkylated with 2-chloroacetamide (55 mM in 100 mM ammonium bicarbonate) and incubated in the dark at ambient temperature for 30 min. The solution was removed and discarded. The gel slices were washed with ammonium bicarbonate (100 mM) for 10 min on a shaker and an equal amount of acetonitrile was added and continued to wash for 10 min on a shaker. The solution was removed, discarded and the gel slices were dried in a speed vac for 45 min. The gel slices were cooled on ice and a cold solution of trypsin (Promega, Madison, WI) 12.5 ng/µL, in ammonium bicarbonate (100 mM) was added, enough to cover the gel slice. After 45 min, the trypsin solution was removed, discarded and an equal amount of ammonium bicarbonate (50 mM) was added and incubated overnight at 37 °C with mixing. Samples were spun down in a microfuge and the supernatants were collected. Peptides were extracted from the gel slices by adding 0.1% trifluoroacetic acid (TFA) enough to cover the slices and mixed at ambient temperature for 30 min. An equal amount of acetonitrile was added and the samples were mixed for an additional 30 min. The samples were spun on a microfuge and the supernatants were pooled. The extract was dried using a speed vac. Samples were desalted using ZipTip C18 (Millipore, Billerica, MA) and eluted with 70% acetonitrile/0.1% TFA. The desalted material was taken to dryness in a speed vac.

Desalted samples were brought up in 2% acetonitrile in 0.1% formic acid and were analyzed by LC/ESI MS/MS with a Thermo Scientific Easy-nLC 1000 (Thermo Scientific, Waltham, MA) nano HPLC system coupled to a tribrid Orbitrap Fusion (Thermo Scientific, Waltham, MA) mass spectrometer. In-line de-salting was accomplished using a reversed-phase trap column (100 µm × 20 mm) packed with Magic C18AQ (5-µm 200Å resin; Michrom Bioresources, Bruker, Billerica, MA) followed by peptide separations on a reversed-phase column (75 µm × 270 mm) packed with Reprosil C18AQ (3-µm 100Å resin; Dr. Maisch, Germany) directly mounted on the electrospray ion source. A 90-minute gradient from 5% to 28% acetonitrile in 0.1% formic acid at a flow rate of 300 nL/minute was used for chromatographic separations. The heated capillary temperature was set to 300 ºC and a static spray voltage of 2200 V was applied to the electrospray tip. The Orbitrap Fusion instrument was operated in the data-dependent mode, switching automatically between MS survey scans in the Orbitrap (AGC target value 500,000, resolution 120,000, and maximum injection time 50 milliseconds) with MS/MS spectra acquisition in the linear ion trap using quadrupole isolation. A 3 second cycle time was selected between master full scans in the Fourier-transform (FT) and the ions selected for fragmentation in the HCD cell by higher-energy collisional dissociation with a normalized collision energy of 27%. Selected ions were dynamically excluded for 20 seconds and exclusion mass by mass width +/- 10 ppm.

Data analysis was performed using Proteome Discoverer 2.5 (Thermo Scientific, San Jose, CA). The data were searched against Uniprot Human (Uniprot UP000005640 March 07, 2021) and cRAP (http://www.thegpm.org/crap/) fasta files. Trypsin was set as the enzyme with maximum missed cleavages set to 2. The precursor ion tolerance was set to 10 ppm and the fragment ion tolerance was set to 0.6 Da. Variable modifications included oxidation on methionine (+15.995 Da), DBCO desthiobiotin AHA on methionine (+712.387 Da) and methionine conversion to AHA (-4.986 Da). Dynamic modifications on the protein N-terminus included acetylation (+42.011 Da), methylation (+14.016 Da) and methionine loss plus acetylation (-89.030 Da). Static modifications included carbamidomethyl on cysteine (+57.021 Da). Data were searched using Sequest HT. All search results were run through Percolator for scoring and identified peptides were filtered for 1% peptide-level false discovery rate using q value of 0.01. LFQ analysis using Minora was performed.

**qPCR assay for KSHV DNA**

Blood and saliva samples were evaluated for KSHV DNA using a quantitative, high-throughput fluorescent-probe-based real-time polymerase chain reaction (PCR) assay (TaqMan assay, Applied Biosystems, Foster City, CA) of the ORF73 gene at the UCI-Fred Hutch Cancer Centre Laboratory in Kampala, Uganda as previously described [5,74]. Samples with >150 copies per mL of KSHV DNA were considered positive [75].

**ELISA**

200 ng/well of gH/gL was adsorbed onto 96 well Immulon 2HB ELISA plates (ThermoFisher Scientific Cat# 3455) overnight at room temperature in a solution of 0.1 M NaHCO$_3$ pH 9.4-9.6. Plates were then washed 4 times with ELISA washing buffer (1 × PBS, 0.02% Tween 20) prior to blocking at 37°C for 1 hour with 250 µl per well of PBS containing 10% Non-Fat Milk and 0.02% Tween 20 (blocking buffer). After blocking, plates were washed 4× with ELISA washing buffer. Serum or plasma was heat-inactivated for 60 minutes at 56°C. Heat-inactivated sserum or plasma was diluted in blocking buffer and three-fold serial dilutions were performed in duplicate followed by a 1 hour incubation at 37°C. Wells coated with gH/gL but without serum/plasma were included to measure background signal. Following 4 additional washes with ELISA washing buffer, a 1:3000 dilution of goat anti-human Ig HRP (Southern Biotech Cat # 2010-05) in blocking buffer was added to each well and incubated at 37°C for 1 hour followed by 4 washes with wash buffer. 50 µl/well of SureBlue Reserve TMB Microwell Peroxidase substrate (SeraCare Cat. #5120–0081) was added. After 3 min, 50 µl/well of 1N Sulfuric Acid was added and the A$_{450}$ of each well was read on a Molecular Devices SpectraMax M2. The binding threshold was defined as the average plus 5 times the SD of the determined by calculating the average of A$_{450}$ values of the control wells. Endpoint titers, defined as the highest dilution of the serum that gives a reading above a predetermined cutoff

based on the average background absorbance of control wells plus 3 standard deviations, were interpolated from the point of the curve that intercepted the binding threshold using the GraphPad Prism 10.0.2 package (Graphpad Software).

## Conjugation of antigens to fluorescently labeled streptavidin

gH/gL was biotinylated at a theoretical 1:1 molar ratio using EZ-Link NHS-PEG4 Biotin (Thermo Fisher Cat# A39259) and the excess was removed using a desalting column. 80 pmol of biotinlyated gH/gL was mixed with Streptavidin-R-phycoerythrin (Agilent Cat# PJRS301-1) (SA-PE) and Streptavidin-allophycocyanin (Agilent Cat# PJ27S-1) (SA-APC) at a 4:1 gH/gL to streptavidin molar ratio respectively. An irrelevant, Avi-tagged biotinylated recombinant decoy protein from *Plasmodium yoelii* (PY-gamma, a kind gift from Dr. D.N. Sather) was conjugated to Streptavidin-R-phycoerythrin labeled with DyLight 650 NHS Ester (Thermo Fisher Cat# 62266) (SA-PE-DL650) to identify cells staining non-specifically with the staining reagent. After 5 min, 25 nmol of free D-Biotin (Invitrogen Cat# B20656) was added to quench any unoccupied SA binding sites.

## B cell sorting

Cryopreserved PBMCs were thawed and resuspended in 200µl of FACS buffer (1X PBS, 2% heat inactivated fetal bovine serum, 1mM EDTA). B cells were isolated using EasySep Human B Cell Enrichment Kit (STEMCELL Technologies Cat# 19054) according to the manufacturer's instructions. Cells were resuspended in 100µl FACS buffer containing mouse serum (Sigma-Aldrich Cat# M5905-5ML), Normal rat serum (STEMCELL Technologies Cat# 13551), unlabeled mouse-anti-Human CD16 (BD Biosciences Cat# 550383), unlabeled mouse-anti-Human CD23 (BD Biosciences Cat# 555707), and unlabeled mouse-anti-Human CD32 (BD Biosciences Cat# 551900) and incubated in the dark at 4°C for 15 min. Cells were diluted to 1ml with FACS buffer, pelleted by centrifugation at 500 x g for 3 min.

1µM of SA-PE and SA-APC conjugated gH/gL were then added to a cocktail antibody panel prepared in 100µl FACS buffer: CD3-FITC (BD Biosciences Cat# 556611) at a 1:100 dilution, CD14-FITC (BD Biosciences Cat# 557153) at a 1:100 dilution, IgD-PerCP-Cy5.5 (eBioscience Cat# 46-9868-42) at a 1:100 dilution, IgG-BV421 (BD Biosciences Cat# 562581, at a 1:200 dilution, CD19-BV711 (BioLegend Cat# 302246) at a 1:200 dilution, and a fixable viability dye eFluor 506 (eBioscience Cat# 65-0866-14) at a 1:200 dilution. 1µM of SA-PE-DL650 conjugated PY-gamma was also added. Cells were incubated for 30 min in the dark at 4°C, and diluted to 1mL with FACS buffer, pelleted by centrifugation at $500 \times g$ for 3 min and then resuspended in 0.5mL of FACS buffer and kept at 4°C. Cells were now subjected to FACS on a BD FACSymphony S6.

Live, antigen-positive class-switched B cells (Live/dead[-], CD3[-], CD14[-], CD19[+], IgD[-], IgG[+], PE-DL650[-], PE[+], APC[+]) were sorted individually into 96 well plates. The cells were then frozen on dry ice and stored at -80°C.

## Amplification and cloning of VH and VL transcripts from sorted cells

RNA from singly sorted cells was reverse transcribed to cDNA by using Superscript IV Reverse Transcriptase (Invitrogen Cat# 18090200) in a 20 µl reaction according to the manufacturer's instructions.

First round reactions contained 5 ul cDNA, 1-unit HotStarTaq Plus DNA polymerase (Qiagen Cat# 203605), 250 nM 3′ primer pool [52], 380 nM 5′ primer pool, [52] 25 mM dNTP (Invitrogen, Cat# 10297018), 2µl 10x buffer, and 12.4µl nuclease-free H$_2$O. Second round PCR reactions used 5 µl first round PCR as template and 250 nM of both 5′ and 3′ primers. Second round PCR primers include homology regions that are corresponding to the leader sequence and constant regions on the expression vector [76]. Second round PCR products were subjected to electrophoresis on a 1% agarose gel containing 0.1% GelRed Nucleic Acid Gel Stain (GoldBio Cat# G-725–500). 10 µL of PCR reactions from wells with appropriately sized amplicons were then purified using Monarch PCR & DNA Cleanup Kit (NEB Cat# T1030L) and eluted in 8 µL DNA Elution Buffer.

Expression plasmid backbones were amplified using primers specific for the leader sequence and constant regions in 25 μL reactions containing 2x Platinum SuperFi II DNA polymerase (Invitrogen Cat# 12368050), 500 nM 5′ and 3′ primers, 10 ng template DNA, and 21.5 μL Nuclease-free water. The reaction was cycled at 98C for 30 s, 30 cycles of 98C for 10 s, 60C for 10 s, and 72C for 3 minutes and 30 s, followed by 72C for 5 minutes. The reaction was treated with 20 units of DpnI (NEB Cat# R0176L) and incubated at 37C for 90 min followed by heat inactivation at 80C for 20 min. The reaction was purified using Monarch PCR & DNA Cleanup Kit according to manufacturer's directions.

The cloning reaction was performed using 20 ng of plasmid backbone, 1 μL of 5x In-Fusion HD Enzyme (Takara Bio Cat# 639650) and the purified second round PCR product for a total reaction volume of 5 ul and incubated at 50ºC for 60 minutes. The assembled products were used to transform DH5 Alpha cells (NEB Cat# C2987H) and plated on agar plates containing ampicillin and incubated overnight at 37ºC. Ampicillin-resistant colonies were used to seed 4 mL LB broth cultures containing 100μg/ml ampicillin. DNA was prepared using QIAprep Spin Miniprep Kit (Qiagen Cat# 27106) and Sanger Sequenced.

## Biolayer interferometry (BLI)

BLI assays were performed on the Octet Red instrument at 30°C with shaking at 1,000 RPM.

**Kinetic analysis.** Anti-Human IgG Fc capture (AHC) sensors (Sartorius Cat# 18–5060) were immersed in kinetics buffer (KB: 1X PBS, 0.01% BSA, 0.02% Tween 20, and 0.005% $NaN_3$, pH 7.4) containing 10 μg/ml of purified mAb for 300s. After loading, the baseline signal was then recorded for 1 min in KB. The sensors were immersed into wells containing serial dilutions of purified recombinant gH/gL for 300s (association phase), followed by immersion in KB for an additional 600s (dissociation phase). The background signal from each analyte-containing well was measured using empty reference sensors and subtracted from the signal obtained with each corresponding ligand-coupled sensor. The background signal of ligand-coupled sensors in KB was subtracted from each sensor at each time-point. Kinetic analyses were performed at least twice with an independently prepared analyte dilution series. Curve fitting was performed using a 1:1 binding model and the ForteBio data analysis software. Mean $k_{on}$ and $k_{off}$ rates were determined by averaging all binding curves that matched the theoretical fit with an $R^2$ value of ≥0.99 and used to calculate the dissociation constant.

**Antibody competition binding assays.** Biotinylated gH/gL was diluted to 290nM and captured on streptavidin biosensors for 300s. The baseline interference was then read for 60s in KB buffer, followed by immersion for 150s (1st association phase) in a 10 ug/mL solution of non-biotinylated mAb to saturate the gH/gL with antibody. The sensors were then dipped in KB for 60s before being immersed in a second well of 10 ug/mL non-biotinylated mAb for 150s (2nd association phase). For MLKH12, mAbs were immersed for 600s rather than 150s to account for slower kinetics of this mAb.

**gH/gL EphA2 binding assays.** Biotinylated EphA2 (Sino Biological Cat: 13926-H27H-B) was diluted to 10 μg/ml and immobilized on streptavidin biosensors for 300s, and then immersed in KB buffer for 60 seconds. Biosensors were then immersed into either KB buffer or into a 100nM solution of gH/gL alone or gH/gL pre-incubated with 200nM of mAb for 300s (association phase) and then into KB for 300s (dissociation phase).

## Virus-free fusion assay

**Seeding.** CHO-K1 cells were seeded in a 6 well dish (Thermofisher Cat. 140675) at 0.3x10^6 cells/well. The volume in each well was brought up to 2 mL using F-12 media (Thermofisher Cat. 11765054) supplemented with 10% FBS, 100U/mL PEN/STREP. Cells were left to grow overnight in a humidified incubator at 37ºC, 5% $CO_2$ (all overnight incubations were done in these conditions).

**Transfection.** The next day, 0.5 ug of KSHV pCAGGS gH, 0.5 ug of KSHV pCAGGS gL, 0.5 ug of EBV pCAGGS gB, and 0.8 ug of T7-Luciferase plasmid were incubated with 7uL GeneJuice (Fisher Scientific Cat. 70967–6) (pre-mixed/ incubated with 250uL of optiMEM media) per well for 15 mins. The GeneJuice:DNA complex was then directly added

dropwise to the plated CHO-K1 cell media. One well was set aside and transfected similarly but replaced the EBV gB pCAGGS plasmid with an empty pTT3 vector backbone as a negative control well. The plate was left overnight at 37C in a humidified incubator.

**Final plating.** The next day, the transfected media was decanted in each well and washed 2 times with sterile PBS (Corning Cat# 21–040-CM) before detaching using 1 mL/well of Gibco Disassociation reagent (Thermo Fisher Scientific Cat# 13151014). The dissociation reagent was incubated for 10 mins at RT and quenched with warmed F-12 media once cells were visibly detached. The detached cells were centrifuged at 500 x g for 3 minutes at RT before being resuspended in 10 mL warm F-12 media. The cells were pelleted and resuspended an additional time before being counted, diluted to 0.3x10^6 cells/mL, and plated at 50uL/well in a 96 well clear bottom TC treated plate (Thermo Fisher Scientific Cat# 07-000-166). 2ug/well of antibodies were added to wells in quadruplicates, 2 sets of wells were left with either no antibody or an irrelevant antibody as positive controls (the former being used as a reference for total fusion). The negative CHO-K1 control transfection well was washed, counted, and plated similarly with no antibody. 293T cells stably expressing the T7 polymerase were also washed, pelleted, and diluted in the same manner using DMEM media (Corning Cat# 15–013-CV) supplemented with 10% FBS, 100U/mL PEN/STREP, 2mM L-glutamine. Using a multichannel pipette, 50uL ($0.3x10^6$ cells/mL) of 293-T7 cells were added to each well. The 96 well plate was then incubated overnight at 37ºC.

**Readout.** On day 4, the 96 well plate was removed and brought to room temperature before the addition of 100µL/well of room temperature SteadyGlo luciferase reagent (Thermo Fisher Scientific Cat# E2520). Each well was mixed thoroughly with a multichannel and incubated for at least 5 minutes before readout on the Glomax Luminometer using the SteadyGlo Luminescence protocol. Negative control wells (containing no EBV gB pCAGGS) were used as a reference for nonspecific luminescence measurement.

**Analysis.** Quadruplicate wells with antibody treatments were averaged and normalized to total fusion luminescence. Total fusion luminescence was determined as the average luminescence readout of the wells with no antibody present. Percent fusion was determined by dividing the average normalized luminescence readout from each antibody treatment by the average total fusion of each plate. At least 3 independent fusion measurements were made for each antibody to compare relative fusion inhibition.

## Negative stain electron microscopy

### KSHV gH/gL and fab complexes preparation for nsEM.

*KSHV gH/gL*: Co-expressed gH/gL proteins were diluted at 0.01 mg/mL in 1x TBS pH 7.4.

*Complex with MLKH1 or MLKH5:* gH/gL protein was complexed with a 2-fold molar excess of MLKH1 Fab and MLKH5 Fab, respectively, and incubated for 30 minutes at room temperature. The complex was diluted to approximately 0.04 and 0.03 mg/mL respectively with 1x TBS pH 7.0.

*gH/gL Complex with MLKH5/MLKH10/MLKH3, MLKH5/MLKH10/MLKH6 or MLKH5/MLKH10/MLKH12*: gH/gL protein was complexed with a 1.2-fold molar excess of MLKH5, a 1.5-fold molar excess of MLKH10, and a 1.2-fold molar excess of either MLKH3 or MLKH6 or a 1.5-fold molar excess of MLKH12, and incubated for 60 minutes at room temperature. The complex was diluted to approximately 0.01 or 0.02 mg/mL with 1x TBS pH 7.0.

*Grid Preparation, Screening, Collection and Processing*: Formvar, stabilized with carbon, 400 mesh, copper ns-EM grids with approximative grid hole size: 42µm (Ted Pella Inc, Cat #01754-F) were glow discharged using the PELCO easi-Glow Glow Discharge Cleaning System (Ted Pella INC, Cat #91000), set to 0.39 mbar with a plasma current of 15mA and a glow process of 30s with 15s hold. 5µl of proteins or protein complexes as described above was allow to settle on a glow discharged grid for 30s, excess liquid on the grids was blotted with filter paper, wash twice with water, once

with 2% (W/V) uranyl formate followed by staining for 30s in 2% uranyl formate, a final step of blotting and air drying of the grids for 3min was done before being stored in a grid box.

Grids were then imaged with the Leginon automated data collection software [77] at 200 keV on a Talos L120C G2 (Thermo Fischer Scientific) using a 4K×4K Thermo Scientific Ceta CMOS Camera at 92,000×magnification, -2 μm defocus, 1.58 Å pixel size, and a total dose of 40 e-/Å². Collected micrographs were transferred in CryoSPARC v4 [78] for processing. Particles were picked using a Gaussian picker (Blob-picker job) and stacked with a box size of 256 pixels with a constant CTF, followed by successive rounds of 2D classification on particles with a visible Fab or shaped matching KSHV structure were picked. Selected 2D classes were used to generate a 3D volume by Ab-Initio Reconstruction, followed by homogeneous refinement and 3D classification. Selected 3D classes were refined and analyzed in UCSF ChimeraX [79] to generate figures. A published gH/gL model (PDB: 7CZF) was used in the structural analysis combined with an AlphaFold 3 [80] prediction of the different Fabs.

### Cryo-electron microscopy

gH/gL was complexed with a 1.2-fold molar excess of MLKH3 Fab, a 1.5-fold molar excess of MLKH10 Fab, and a 1.2-fold molar excess of MLKH5 Fab, and incubated for 60 minutes at room temperature. The complex was then concentrated to 5 mg/ml, flash frozen and stored at -80°C until further use. The complex was then diluted to approximately 1 mg/mL with 1x TBS pH 7.0 and 3μl was loaded onto Quantifoil R1.2/R1.3 + 2mm Cu300 mesh grids (Ted Pella) for non-tilted acquisition and onto UltrAuFoil R1.2/1.3 Au300 mesh grids (Ted Pella) for 30° tilted acquisition due to the observed preferred orientation during screening. A Tergeo-Plus plasma cleaner (PIE scientific, Cat#PIE-TP100) was used to discharge the grids. Grids were vitrified on a Vitrobot Mark IV (ThermoFisher) with a blot force of 0, blot time of 5 sec and wait time of 0 sec between application for Quantifoil grids and with a blot force of 1, blot time of 5 sec and wait time of 3 sec between application for UltrAuFoil grids. Blotting was done at 4°C and 100% humidity before plunging into liquid ethane.

Data was collected on a 200 kV Glacios microscope (ThermoFisher) with a K3 direct electron detector (Gatan) at a magnification of 36,000x (1.122 Å per pixel) using SerialEM [81]. A total of 5271 movies were collected non-tilted with the Quantifoil grids and 7449 movies were collected with a 30° tilt for the UltrAuFoil grids. Motion correction was performed using Warp [82] and processed using cryoSPARC [78] (V4.7.1). After contrast transfer function estimation [83], curate exposure and automated blob picking, initial 2D templates were generated through multiple rounds of 2D classification with a particle down sampling to a specified box size, using Fourier cropping to fasten the process. Using final 2D classes, particles were re-extracted without cropping. A total of ~11 million and ~9 million particles respectively were extracted and after 6 and 4 rounds of 2D classification, ~1 million particles were merged together and used for Ab-initio reconstruction without symmetry and 3000 initial iterations. The reconstruction was refined using non-uniform refinement [84] with C1 symmetry followed by a round of 3D classification with 10 classes, 6Å filter resolution, 4 O-EM epochs of 10,000 particles per class with a learning rate of 0.5, The convergence criterion was set to 0.001% and F-EM of 50. 2 main classes were found. One of these classes was subject to an Ab-initio reconstruction then new 2D classification rounds were performed to ensure the absence of junk particles. A rebalance job was performed to clear any preferential orientation with a rebalance percentile of 80. Following these steps, the remaining 75,257 particles were used in Ab-initio reconstruction without symmetry followed by a non-uniform refinement in C1. The resolution was estimated to be 3.51 Å per the gold standard FSC 0.143. Local resolution was estimated based on gold standard FSC 0.143 and is shown in S3F-S3G Fig.

The model of gHgL heterodimer from PDB ID 7CZF [45] was used as the initial model and a model for each Fab was generated by AlphaFold3 [80]. Both models were fitted into the cryo-EM map using ChimeraX [79] (v1.9), only the Fv protein of each Fab was used. MLKH5 was not built in because of its lower resolution (~4.5 Å). Model building was completed

using Coot [85] (v0.9.8.95 EL and 1.1.18) and refinement was run using Phenix (v2.0.5793) real_space_refinement [86]. Further refinement was performed using ISOLDE [87] in ChimeraX and Coot. Data collection and refinement statistics are summarized in S3 Table. Structural figures were generated using ChimeraX. BSA data was determined using the PDBePISA server [88] and graphs were created using GraphPad Prism 10 for Windows, GraphPad Software, Boston, Massachusetts USA.

### Biotinylation

Freshly thawed KSHV gH/gL HisAvi was biotinylated with EZ-Link NEH PEG4 biotin (10:1 molar ratio) (Thermo Fisher Scientific *Cat# A39259*) overnight at 4ºC with rotation. Biotinylated gH/gL was then purified via desalting column (Thermo Fisher Scientific *Cat# 87769*) to remove remaining free biotin according to manufacturer protocol using 1X PBS.

### Neutralization assays

**Participant sera on vero cells.** Neutralization assays with participant sera in epithelial cells were performed using Vero cells (ATCC CCL-81). Vero cells were seeded at 12,500 cells/well in flat-bottom 96-well plates, 48 hours before each assay. 16 μL of heat-inactivated sera diluted in 64 μL of cDMEM was incubated with 80ul of rKSHV.219 (produced from BAC16-iSLK cells) diluted to achieve 100 plaque forming units (pfu)/well in control wells without sera for 25 minutes at 37°C. Vero cells were then incubated with 50 μL of the sera/virus mixture and 2.67 μL of polybrene (0.25mg/mL) (Sigma Aldrich, Cat. TR-1003) for 1 hour at 37°C. After incubation, 100 μL of cDMEM was added to each well. The final dilution of each patient sample is reported as the dilution prior to the addition of 100ul of cDMEM. Five days later, fluorescent plaques were detected with a Typhoon Trio imager and counted in ImageJ. All samples and controls were run in triplicate and averaged.

**Monoclonal antibodies on vero cells.** Neutralization assays with monoclonal antibodies in epithelial cells were performed using Vero cells. Vero cells were seeded at 12,500 cells/well in flat-bottom 96-well plates, 48 hours before each assay. Monoclonal antibodies were serially diluted in 80 μL of cDMEM. Diluted antibody was combined with 60 uL of KSHV diluted to achieve 100 pfu/well in antibody negative control wells for 25 minutes at 37°C. This mixture was incubated for 25 minutes at 37°C. Vero cells were then incubated for 1 hour at 37°C with 50 μL of the virus/mAb mixture and 2.67 uL of polybrene (0.25 mg/mL). After incubation, 100 μL of cDMEM was added to each well. The concentration of each mAb is reported as the dilution prior to the addition of 100ul of cDMEM. Five days post-infection, fluorescent plaques were detected with a Typhoon Trio imager and counted in ImageJ. All samples and controls were run in duplicate and averaged within an assay. Each assay was repeated at least 2 times.

**Monoclonal antibodies on RPMI8226 cells.** Neutralization assays with monoclonal antibodies in B cells were performed using RPMI8226 cells (ATCC CCL-155). RPMI cells were diluted to 100,000 cells/ml in cRPMI. 52ul of cells were added to each well in flat-bottom 96-well plates at the beginning of the assay. Monoclonal antibodies and KSHV were both diluted in the same media as the cells. Each mAb was diluted to a concentration of 444 μg/ml in 50ul of cDMEM and mixed with 50 ul of KSHV, diluted to achieve a 5% infection rate, and incubated for 25 minutes at 37°C. 45 μl of the antibody/virus mixture was added to the RPMI cells along with 2.67ul of polybrene (0.25mg/ml) before being centrifuged at 1,500 X *g* for 30 minutes. Three days post-infection, cells were fixed with 2% paraformaldehyde (Electron Microscopy Sciences, Cat. 15714- S) for 18 minutes then resuspended in 1xPBS (Gibco, Cat. 14190–144) supplemented with 2% FBS. GFP was analyzed on a BD FACSymphony A5. All samples and controls were run in duplicate and averaged.

### Supporting information

**S1 Table. Ontogenies of gH/gL specific antibodies.**
(DOCX)

**S2 Table. Properties of gH/gL specific antibodies.**
(DOCX)

**S3 Table. Data collection and refinement statistics for Cryo-EM structure statistics.**
(TIF)

**S1 Fig. Peptide coverage of gH/gL detected by mass spectrometry.** The indicated bands from the right-hand lane of the SDS-PAGE gel in Fig 1D were excised an analyzed by proteolytic cleavage followed by mass spectrometry. The peptides identified in the mass spectra that map to gH (**A**) and gL (**B** and **C**) are highlighted on the protein sequences in green. Cysteine residues with carbamidomethylation and oxidized methionine residues are indicated with C and O, respectively.
(TIF)

**S2 Fig. Plasma and serum reactivity to gH/gL in study participants.** Source data from Fig 2A. (**A**) Plasma from mothers who tested positive for KSHV DNA (+), (**B**) Plasma from household children who tested positive for KSHV DNA (+), and (**C**) plasma from household children who tested negative for KSHV DNA (-), as well as serum from presumed KSHV-negative donors from the Seattle-area were serially diluted and tested for binding to gH/gL by ELISA as indicated.
(TIF)

**S3 Fig. Cryo-EM processing workflow for MLKH5/MLKH3/MLKH10 complexed with gH/gL. (A)** Data processing workflow (**B**) 2D classification. (**C**) View direction distribution plot of particles in 3D reconstruction. (**D**) Resolution estimation, CryoSPARC GSFSC. (**E**) Model fit image to 3D reconstruction, focused on MLKH10 and gH interface. (**F-G**) Local resolution map emphasizing the MLKH10 interface (**F**) and a 90° rotated view for the MLKH3 interface (**G**).
(TIF)

**S4 Fig. In depth views of the different CDR regions for MLKH3 and MLKH10.** (**A-C**) MLKH3 CDR-H1, H2 and H3 are shown in teal with gH in grey surface as indicated. (**D-F**) MLKH10 CRD-H1, H2 and H3 shown in orange with gH in grey surface and gL in yellow surface. (**G-I**) MLKH10 CRDR-L1, L2 and L3 shown in salmon with gH in grey surface and gL in yellow surface. In all panels hydrogen bonds are shown as cyan dashed lines and salt bridges as yellow dashed lines.
(TIF)

## Acknowledgments

We acknowledge the use of instruments acquired through other funding mechanisms including the M. J. Murdock Charitable Trust and the NIH S10 Shared Instrumentation Grant.

## Author contributions

**Conceptualization:** Warren Phipps, Marie Pancera, Jim Boonyaratanakornkit, Andrew T. McGuire.

**Formal analysis:** Yu-Hsin Wan, Nicholas T. Aldridge, Kevin Lang, Holly M. Dudley, Gargi Kher, Andrew T. McGuire.

**Funding acquisition:** Warren Phipps, Jim Boonyaratanakornkit, Andrew T. McGuire.

**Investigation:** Yu-Hsin Wan, Sara Pernikoff, Nicholas T. Aldridge, Kevin Lang, Holly M. Dudley, Samuel C. Scharffenberger, Gargi Kher.

**Resources:** Warren Phipps, Marie Pancera, Jim Boonyaratanakornkit, Andrew T. McGuire.

**Visualization:** Yu-Hsin Wan, Sara Pernikoff, Nicholas T. Aldridge, Kevin Lang, Holly M. Dudley, Samuel C. Scharffenberger, Gargi Kher.

**Writing – original draft:** Holly M. Dudley, Andrew T. McGuire.

**Writing – review & editing:** Yu-Hsin Wan, Sara Pernikoff, Nicholas T. Aldridge, Kevin Lang, Holly M. Dudley, Samuel C. Scharffenberger, Gargi Kher, Warren Phipps, Marie Pancera, Jim Boonyaratanakornkit, Andrew T. McGuire.

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
