## [Decision Letter · Decision Letter 0]

9 Oct 2025

PPATHOGENS-D-25-02243

Monoclonal neutralizing antibodies elicited by infection with Kaposi sarcoma-associated herpesvirus reveal critical sites of vulnerability on gH/gL

PLOS Pathogens

Dear Dr. McGuire,

Thank you for submitting your manuscript to PLOS Pathogens. After careful consideration, we feel that it has merit but does not fully meet PLOS Pathogens's publication criteria as it currently stands. Therefore, we invite you to submit a revised version of the manuscript that addresses the points raised during the review process.

Please submit your revised manuscript within 30 days Dec 07 2025 11:59PM. If you will need more time than this to complete your revisions, please reply to this message or contact the journal office at plospathogens@plos.org. Please include the following items when submitting your revised manuscript:

We look forward to receiving your revised manuscript.

Kind regards,

Richard James Stanton

Academic Editor

PLOS Pathogens

Alison McBride

Section Editor

PLOS Pathogens

Sumita Bhaduri-McIntosh

Editor-in-Chief

PLOS Pathogens

orcid.org/0000-0003-2946-9497

Michael Malim

Editor-in-Chief

PLOS Pathogens

orcid.org/0000-0002-7699-2064

**Journal Requirements:**

At this stage, the following Authors/Authors require contributions: Yu-Hsin Wan, Sara Pernikoff, Nicholas Aldridge, Kevin Lang, Holly Dudley, Samuel Scharffenberger, Gargi Kher, Warren Phipps, Marie Pancera, Jim Boonyaratanakornkit, and Andrew McGuire. Please ensure that the full contributions of each author are acknowledged in the "Add/Edit/Remove Authors" section of our submission form.

https://journals.plos.org/plospathogens/s/submission-guidelines#loc-parts-of-a-submission

4) We noticed that you used the phrase 'data not shown' in the manuscript. We do not allow these references, as the PLOS data access policy requires that all data be either published with the manuscript or made available in a publicly accessible database. Please amend the supplementary material to include the referenced data or remove the references.

5) We do not publish any copyright or trademark symbols that usually accompany proprietary names, eg ©,  ®, or TM  (e.g. next to drug or reagent names). Therefore please remove all instances of trademark/copyright symbols throughout the text, including:

- TM on page: 31.

6) Please upload all main figures as separate Figure files in .tif or .eps format. For more information about how to convert and format your figure files please see our guidelines: 

7) We notice that your supplementary Figure, and Table are included in the manuscript file. Please remove them and upload them with the file type 'Supporting Information'. Please ensure that each Supporting Information file has a legend listed in the manuscript after the references list.

8) Some material included in your submission may be copyrighted. According to PLOSu2019s copyright policy, authors who use figures or other material (e.g., graphics, clipart, maps) from another author or copyright holder must demonstrate or obtain permission to publish this material under the Creative Commons Attribution 4.0 International (CC BY 4.0) License used by PLOS journals. Please closely review the details of PLOSu2019s copyright requirements here: PLOS Licenses and Copyright. If you need to request permissions from a copyright holder, you may use PLOS's Copyright Content Permission form.

Potential Copyright Issues:

i) Figures 1A: We noted that the figure is created through BioRender. Please confirm that you hold a Premium account and provide a pdf copy of the CC BY 4.0 Licence as provided by BioRender. For instructions on how to generate a CC BY 4.0 license for your figure, please see the guidelines here: https://help.biorender.com/hc/en-gb/articles/21282341238045-Publishing-in-open-access-resources. 

If you are using the free assets from BioRender, we are unable to publish these images as they are licenced under a stricter licence than CC BY 4.0. In this case we ask you to remove the BioRender images and replace them with open source alternatives.

See these open source resources you may use to replace images / clip-art:

- https://bioart.niaid.nih.gov/ 

- https://bioicons.com/

- https://healthicons.org/ 

- https://scidraw.io/

- https://reactome.org/icon-lib

ii) Figure 3A. Please confirm whether you drew the images / clip-art within the figure panels by hand. If you did not draw the images, please provide (a) a link to the source of the images or icons and their license / terms of use; or (b) written permission from the copyright holder to publish the images or icons under our CC BY 4.0 license. Alternatively, you may replace the images with open source alternatives. See these open source resources you may use to replace images / clip-art:

9) Thank you for stating that " Negative-staining EM maps are deposited in the EMDB with accession codes: EMD-72129 (Negative Stain EM map of KSHV glycoprotein gH and gL), EMD-72130 Negative Stain EM map of KSHV glycoprotein gH/gL in complex with MLKH1 Fab, EMD-72131Negative Stain EM map of KSHV glycoprotein gH/gL in complex with MLKH5 Fab), EMD-72132 Negative Stain EM map of KSHV glycoprotein gH/gL in complex with MLKH5, MLKH10 and MLKH3 Fabs. EMD-72133 Negative Stain EM map of KSHV glycoprotein gH/gL in complex with MLKH5, MLKH10 and MLKH6 Fabs. EMD-72525 Negative Stain EM map of KSHV glycoprotein gH/gL in complex with MLKH5, MLKH10 and MLKH12 Fabs. The seuqences of the antibodies described herein are available on NCBI Genbank; accession numbers: PV035824-PV035843."  Please note that, though access restrictions are acceptable now, your entire minimal dataset will need to be made freely accessible if your manuscript is accepted for publication. This policy applies to all data except where public deposition would breach compliance with the protocol approved by your research ethics board.

10) In the online submission form, you indicated that "Requests for resources and reagents should be directed to and will be fulfilled by Andrew T. McGuire (amcguire@fredhutch.org). All materials generated herein are available upon request under an MTA from the corresponding author (amcguire@fredhutch.org). The pTT3 vectors are used under license from the National Research Council of Canada. Any additional information required to reanalyze the data reported in this paper is available from the lead contact upon request." All PLOS journals now require all data underlying the findings described in their manuscript to be freely available to other researchers, either

1. In a public repository

2. Within the manuscript itself

3. Uploaded as supplementary information.

11) Please amend your detailed Financial Disclosure statement. This is published with the article. It must therefore be completed in full sentences and contain the exact wording you wish to be published.

2) State what role the funders took in the study. If the funders had no role in your study, please state: "The funders had no role in study design, data collection and analysis, decision to publish, or preparation of the manuscript.".

Note: Please ensure that the funders and grant numbers match between the Financial Disclosure field and the Funding Information tab in your submission form. Note that the funders must be provided in the same order in both places as well.

**Reviewers' Comments:**

Reviewer's Responses to Questions

**Part I - Summary**

Reviewer #1: Wu et al. report a characterization of a panel of human antibodies targeting the gH/gL glycoprotein complex of KSHV, an oncogenic human virus. This work addresses an important gap in knowledge, as previous studies have been limited to the use of polyclonal sera (animal or human). In this manuscript, a dozen human mAbs, isolated from KSHV-infected individuals, are characterized in terms of their binding affinity and their potency in neutralizing the virus or blocking cell-cell fusion. Low-resolution reconstructions of five neutralizing mAbs bound to gH/gL, obtained from negative-stain EM images, are provided, revealing vulnerable sites on the viral glycoprotein. The data are, in general, well-presented, and the experiments are well-designed and executed, with some room for improvement as suggested in the major and minor points sections below.

Reviewer #2: Wan et al. report highly interesting findings on the discovery of neutralizin antibodies to KSHV gH/gL and distinct antigenic sites on KSHV gH/gL that serve as targets for these neutralizing antibodies. The authors used a cohort of Ugandan mothers and children to determine seroreactivity to KSHV gH/gL and to isolate neutralizing antibodies through antigen-specific B cell sorting. The finding that most subjects in the cohort show high reactivity to gH/gL is interesting, there is a surprisingly high number of potent neutralizers among the mothers and also their children, which conflicts with the literature. This part of the study raises some questions that are not properly discussed. Also, the KSHV PCR is not properly explained (RT-PCR?). Showing some true negative controls would be really helpful here, and the neutralization assay might suffer from complement effects if the methods are as stated in the manuscript and did not incorporate a heat inactivation step.

The identification of 12 binding and neutralizing antibodies is of high value to the community.

In their structural and elegant mutual competition studies, in addition to the EphA2 receptor binding site, the authors identified two additional antigenic sites as targets of neutralizing antibodies. Through electron microscopy analysis, the authors identified the rough location of the antigenic sites.

Sequence analysis showed that some antibodies from one individual were clonally related or shared VH/VL gene usage, which may suggest reproducible pathways of antibody development.

The functional charactization of the antibodies is interesting, in particular the observed differences between neutralizing acitivty in an epithelial cell infection assay, a cell-cell-based fusion assay, and an infection assay with a B cell line. The latter findings suffer from some overinterpratation.

Whether the structural nsEM data is sound/of sufficient quality or needs further experimental confirmation through e.g. mutagenesis should be decided by an expert reviewer with deeper understanding of these technical aspects.

**Part II – Major Issues: Key Experiments Required for Acceptance**

Reviewer #1: • Lines 119-22 and Fig 1D and 1E: the 3D reconstruction of gH/gL does not seem convincing. The ribbon model (1E) should be placed inside the map (1D), and the correlation coefficient (cc) should be provided. Otherwise, the panels D and E may be the best removed, as they do not add to the story and indeed gH/gL may be too small to obtain good 3D reconstructions. It would also be helpful if gH and gL were represented in more contrasted colors (thru the manuscript, figures 6D, 7 included).

• CC values for fitting of the Fabs and gH/gL into maps should be provided for representations on Fig 6B, 6C and Fig 7.

• Hahn et al. have shown that KSHV gH can be expressed and secreted without gL, which typically plays the role of a chaperone and is essential for gH expression in other herpesviruses (PMID: 18945775). In this manuscript, the gH/gL complex was purified via an affinity tag at the C-terminus of gH, which likely resulted in a SEC peak containing gH alone as well as the gH/gL complex. Thus, the statement in lines 125-126, "gH/gL is highly pure," is questionable. Could the authors comment on that? Were there indications of particles lacking one domain (i.e., gL) in the 2D classes?

• Have the authors run AlphaFold multimer on gH/gL complexes with individual Fabs? This would be interesting and complementary information to the nsEM structural models reported in the manuscript. pTM and ipTM scores should be included to indicate the quality / confidence of the AF predicted models.

Reviewer #2: Unclear whether necessary: If the data can be interpreted in a more careful way, the binding and neutralization experiments are good enough as is, otherwise some negative controls would be needed. If neutralization was done without heat inactivation, then a repeat with heat inactivation is needed to make statements about neutralization in Fig 2.

**Part III – Minor Issues: Editorial and Data Presentation Modifications**

Reviewer #1: • Fig 1 – D to F panels are not labelled

• Line 57: consider including PMID: 38844783, PMID: 19536280 with reference 17

• Line 80: references 34-38 do not all relate to the RGD motif being the binding site on integrins. The references should be disperesed throughout the sentence to reflect corresctly what they refer to

• Line 86: consider adding PMID: 34499637 to reference 44

• Line 86: typo in “crystallography’

• Line 103: “12 high-affinity monoclonal …”- high-affinity typically refers to Kd values in nM range, which was not the case for all 12 mAbs. Consider rephrasing.

• Line 103: “…map to 5 distinct epitope regions”- shouldn’t it be 4? (Figure 7D)

• Line 140: Could the authors clarify this statement? Was the blood collected and tested (Figure 2) prior to the RT-PCR that was used to assess the KSHV infection status?

• Lines 149 – 151: Could the authors hypolesize why those two samples (009 and 039-C1) could reduce infectivity by more than 50%?

• Lines 163-172: If understood correctly, the sera from 3 KSHV+ and 1 KSHV- participants were used for isolation of B-cells. Does Figure 3B show results for one of the participants? It should be stated which one it was, because it seems that the majority of gH/gL specific B-cells (9 shown on Figure 3B) came from a single participant?

• Line 195: “… map to 6 epitope clusters…”Figure 4 shows presence of 5 clusters (the same was mentioned in the abstract, line 25).

• Line 289: It is confusing, but ephrin is the name for the ligand for Eph receptors. Therefore, ‘ephrin’ should be replaced with ‘Eph’

• Line 301: ‘intimates’was probably meant to be ‘indicates’?

• Lines 303-304: Could the authors speculate why competing mAbs do not have the same neutralization potency?

• Line 311: an extra letter e is present

• Line 319: ‘ephrin A2’ should be replaced with ‘EphA2’

• Line 358, 551, 676: track changes kept on

• Line 475: a space needed after the closed bracket

• Line 502: a definition of ‘Endpoint titer’ could be helpful for non-experts

• Lines 517-524: the word mouse should not be capitalized

• Lines 663-665: proof-reading is needed to correct wrongly capitalized words (in figure legends as well)

• Line 697: the virus full-name (strain) should be included

• Line 709: extra .

• Line 712: mAb instead of mab

• Line 719: word ‘determined’ should be replaced with a better one

• Line 721: appoximating to 52ul of cells would be reasonable here

• References, lines 731-1014 are redundant (and missed a number somewhere) to the ones in lines 1110-1400, which seem to be correct.

• A table summarizing the findings would be helpful to present all the data in one place. For each mAb, there could be columns containing the information on human participant, binding affinity, assigned cluster, neutralization potency, ability to block cell-cell fusion, having AF model and / or experimental 3D reconstruction, GenBank number etc

• Figure 2A: consider marking the 2 participants whose PMBCs were used for B-cell sequencing. Instead of ‘PCR+’ and ‘PCR-‘ labels, consider using ‘KSHV+’ and ‘KSHV-‘.

• Line 1036: define what ‘C1’ refers to.

• Figure 3A: what are the 4 blue balls in the center representing?

• Figure 5I: is there a special reason why the order of the mAbs (x axis) is not the same as on panels A, J and K?

• Figure 7D: could Fabs MLKH6 and 12 be represented in different shades of red (or put one in grey and one in red) to distinguish them on the figure? Figure 7 in general is important, and merits a better representation of structures (for ex. maps more transparent, ribbon models more visible, panels larger)

• Line 1100: ‘…fitted into the ‘densities’. Strictly speaking, cryo-EM maps should not be referred to as densities but as Coulomb potential maps (PMID: 31400843).

Reviewer #2: - Line 89: “binding“ is missing.

- Line 135, line 418: Please explain RT-PCR. Why was reverse-transcriptase PCR used for KSHV diagnostics? What was the target? Where is the primer(probe?) set described with regard to the target? Ths information is missing from the manuscript.

- The authors constantly use either RMPI8226 or RPMI8226 throughout the text? Which is correct? Reference 55 is not easily accessible, RPMI8226 points to RRID:CVCL_0014 and makes sense.

- Fig 2 A and line 501: As per S1 Fig, there is a lot of binding to gH/gL below 1:100 dilution with the negative control serum. In fact, almost all PCR-negative children show endpoint titers below 100. This is a bit suspicious, could they in fact be negative? It is unclear from the methods how the threshold for the binding ELISA was defined (what were the control wells?). This needs to be made clear. Again, some negative controls might help here.

- Fig 2: Showing some controls would illustrative in particular for B, but also A. Were there any truly, or with high certainty, negative sera included? Other studies report rather low neutralization at 1:100, e.g. Byren et al, JGV 2024- At 1:50, Mortazavi et al, Viruses 2020 report similar neutralization, albeit with mostly KS patient seraa, and also not showing a negative control. Here, 1:10 was used, which is a very low dilution.

- There is no mention of heat inactivation fort he neutralization assay – this would be critical to distinguish complement-mediated effects from true neutralization. At 1:10, complement activation through binding antibodies will override any neutralization effects. It would also explain the high proportion of sera with neutralizing activity, also in PCR-negative children.

- Along those same lines: The overall fact that the mothers cohort universally has highly neutralizing serum activity is a bit suspicious compared to the literature. Kumar et al, PLoS One 2023, found that less than 10% of of asymptomatically infected individuals in a cohort from Zambia had neutralizing antibodies.

- Line 243: “Collectively these data indicate that with the exception of MLKH 10 (Cluster 5), at least one mAb from each cluster can neutralize KSHV infection of epithelial cells, but none can neutralize KSHV infection of B cells.“ This is quite an overinterpretation. The antibodies can’t neutralize KSHV infection of RPMI8226 cells, that’s it.

PLOS authors have the option to publish the peer review history of their article (what does this mean? ). If published, this will include your full peer review and any attached files.

**Do you want your identity to be public for this peer review?** For information about this choice, including consent withdrawal, please see our Privacy Policy .

Reviewer #1: No

Reviewer #2: No

**Figure resubmission:**
---

## [Decision Letter · Decision Letter 1]

29 Nov 2025

Dear Dr. McGuire,

We are pleased to inform you that your manuscript 'Monoclonal neutralizing antibodies elicited by infection with Kaposi sarcoma-associated herpesvirus reveal critical sites of vulnerability on gH/gL' has been provisionally accepted for publication in PLOS Pathogens.

Best regards,

Richard James Stanton

Academic Editor

PLOS Pathogens

Alison McBride

Section Editor

PLOS Pathogens

Sumita Bhaduri-McIntosh

Editor-in-Chief

PLOS Pathogens

orcid.org/0000-0003-2946-9497

Michael Malim

Editor-in-Chief

PLOS Pathogens

orcid.org/0000-0002-7699-2064

Reviewer Comments (if any, and for reference):

Reviewer's Responses to Questions

**Part I - Summary**

Reviewer #1: The authors have well addressed all the remarks I had. The revised manuscript is greatly improved and will be of interest to the whole herpesvirus community. Congratulations to the authors!

Reviewer #2: The authors addressed all issues that were raised. It is commendable that they updated the structural data. This is a very nice and interesting study.

**Part II – Major Issues: Key Experiments Required for Acceptance**

Reviewer #1: (No Response)

Reviewer #2: (No Response)

**Part III – Minor Issues: Editorial and Data Presentation Modifications**

Reviewer #1: I would suggest adding a sentence in discussion, that AF was performed but that the predicted models had low ipTM values and did not resemble experimentally determined structures. This is a good argument for why the structures in this report, even though at low resolution, are relevant and why relying on AF models would have been misleading.

Reviewer #2: (No Response)

PLOS authors have the option to publish the peer review history of their article (what does this mean? ). If published, this will include your full peer review and any attached files.

**Do you want your identity to be public for this peer review?** For information about this choice, including consent withdrawal, please see our Privacy Policy .

Reviewer #1: No

Reviewer #2: No

---

## [Editor Report · Acceptance letter]

Dear Dr. McGuire,

We are delighted to inform you that your manuscript, " 

Monoclonal neutralizing antibodies elicited by infection with Kaposi sarcoma-associated herpesvirus reveal critical sites of vulnerability on gH/gL ," has been formally accepted for publication in PLOS Pathogens.

Best regards,

Sumita Bhaduri-McIntosh

Editor-in-Chief

PLOS Pathogens

orcid.org/0000-0003-2946-9497

Michael Malim

Editor-in-Chief

PLOS Pathogens

orcid.org/0000-0002-7699-2064